# Global Normalization for Streaming Speech Recognition in a Modular Framework

**Ehsan Variani, Ke Wu, Michael Riley, David Rybach, Matt Shannon, Cyril Allauzen**

Google Research
{variani,wuke,riley,rybach,mattshannon,allauzen}@google.com

## Abstract

We introduce the Globally Normalized Autoregressive Transducer (GNAT) for addressing the label bias problem in streaming speech recognition. Our solution admits a tractable exact computation of the denominator for the sequence-level normalization. Through theoretical and empirical results, we demonstrate that by switching to a globally normalized model, the word error rate gap between streaming and non-streaming speech-recognition models can be greatly reduced (by more than 50% on the Librispeech dataset). This model is developed in a modular framework which encompasses all the common neural speech recognition models. The modularity of this framework enables controlled comparison of modelling choices and creation of new models. A JAX implementation of our models has been open sourced.[1]

## 1  Introduction

Deep neural network models have been tremendously successful in the field of automatic speech recognition (ASR). Several different models have been proposed over the years: cross-entropy (CE) models with a deep feed-forward architecture [19], connectionist temporal classification (CTC) models [14] with recurrent architectures such as long short-term memory (LSTM) [20], and more recently sequence-to-sequence (Seq2Seq) models like listen, attend and spell (LAS) [8], recurrent neural network transducer (RNN-T) [13], and hybrid autoregressive transducer (HAT) [40]. When configured in non-streaming mode, these neural ASR models have reached state-of-the-art word error rate (WER) on many tasks. However, the WER significantly worsens when they are operating in streaming mode. In this paper, we argue that one main cause of this WER gap is that all the existing models are constrained to be locally normalized which makes them susceptible to the *label bias* problem [36, 2, 24, 4]. To address this problem, we introduce new category of globally normalized models called *Globally Normalized Autoregressive Transducer* (GNAT). Our contributions are:

(1) **Addressing the label bias problem in streaming ASR through global normalization** that significantly closes more than 50% of the WER gap between streaming and non-streaming ASR.
(2) **Efficient, accelerator-friendly algorithms for the exact computation of the global normalization** under the finite context assumption.
(3) **A modular framework for neural ASR** which encompasses all the common models (CE, CTC, LAS, RNN-T, HAT), allowing creation of new ones, and extension to their globally normalized counterparts.

---

[1] https://github.com/google-research/last

36th Conference on Neural Information Processing Systems (NeurIPS 2022).

## 2 Streaming Speech Recognition

For an *input feature sequence* $\mathbf{x} = x_1 \ldots x_T$, usually represented as a sequence of real valued feature vectors (such as log mel), and a finite output alphabet $\Sigma$, we wish to predict the corresponding *output label sequence* $\mathbf{y} = y_1 \ldots y_U$, $y_i \in \Sigma$. We call each element $x_i$ in $\mathbf{x}$ a *frame*, and each element $y_i$ in $\mathbf{y}$ an *output label*. Common ASR models do not directly predict $\mathbf{y}$, but rather an *alignment label sequence* $\mathbf{z} = z_1 \ldots z_V$, $z_i \in \Sigma \cup \Delta$. $\Delta$ is a finite alphabet of control labels, such as the blank label in CTC or RNN-T, or the end-of-sequence label in LAS. There is a deterministic mapping $B(\mathbf{z}) : (\Sigma \cup \Delta)^* \to \Sigma^*$ for obtaining $\mathbf{y}$ from $\mathbf{z}$ (e.g. in RNN-T, we simply remove all the blank labels from $\mathbf{z}$). An ASR model can then be broken down in two tasks: (1) assigning a score $\prod_i \omega(z_i | \mathbf{x}, \mathbf{z}_{<i})$ to each alignment sequence $\mathbf{z}$, where $\omega(z_i | \mathbf{x}, \mathbf{z}_{<i}) > 0$ is the *alignment score* of predicting a single alignment label; (2) finding (usually approximately) $\arg\max_{\mathbf{y}} \sum_{\mathbf{z} | B(\mathbf{z}) = \mathbf{y}} \prod_i \omega(z_i | \mathbf{x}, \mathbf{z}_{<i})$.

A non-streaming ASR model's alignment score $\omega(z_i | \mathbf{x}, \mathbf{z}_{<i})$ has access to the entire $\mathbf{x}$ for any $i$. In contrast, a streaming model's alignment score takes the form $\omega(z_i | \mathbf{x}_{<t(i)}, \mathbf{z}_{<i})$: it only has access a prefix $\mathbf{x}_{<t(i)}$ of the input feature sequence, where $t(i)$ is the frame to which $z_i$ is *aligned*. Streaming models can be seen as special case of non-streaming models with respect to the alignment scores.

All the common neural ASR models use a locally normalized alignment score which satisfies the constraint $\sum_{z \in \Sigma \cup \Delta} \omega(z_i = z | \mathbf{x}, \mathbf{z}_{<i}) = 1$. This is achieved by applying the softmax function to the last layer activations. The local normalization constraint makes $\omega(z_i | \mathbf{x}, \mathbf{z}_{<i})$ easily interpretable as a conditional probability distribution $P_\omega(z_i | \mathbf{x}, \mathbf{z}_{<i})$, and thus $\prod_i \omega(z_i | \mathbf{x}, \mathbf{z}_{<i})$ easily interpretable as $P_\omega(\mathbf{z} | \mathbf{x})$. With samples from the true data distribution $P(\mathbf{x}, \mathbf{y})$, the modeling parameters are optimized by minimizing the negative log-conditional-likelihood loss $E_{P(\mathbf{x}, \mathbf{y})}[- \log P_\omega(\mathbf{y} | \mathbf{x})] = E_{P(\mathbf{x}, \mathbf{y})}[- \log \sum_{\mathbf{z} | B(\mathbf{z}) = \mathbf{y}} P_\omega(\mathbf{z} | \mathbf{x})]$.

### 2.1 Label Bias in Streaming ASR

For a non-streaming, locally normalized model, the negative log-conditional-likelihood loss is minimized *in the limit* (assuming enough model capacity) by setting $\omega(z_i | \mathbf{x}, \mathbf{z}_{<i})$ to the true conditional probability $P(z_i | \mathbf{x}, \mathbf{z}_{<i})$, leading to $\prod_i \omega(z_i | \mathbf{x}, \mathbf{z}_{<i})$ being equal to the true posterior $P(\mathbf{z} | \mathbf{x})$.

For a streaming model, $\mathbf{x}$ is replaced by $\mathbf{x}_{<t(i)}$ in the alignment score (e.g. by using a unidirectional encoder). Here the negative log-conditional-likelihood loss is minimized *in the limit* by setting $\omega(z_i | \mathbf{x}_{<t(i)}, \mathbf{z}_{<i})$ to $P(z_i | \mathbf{x}_{<t(i)}, \mathbf{z}_{<i})$. As a result, the product $\prod_i \omega(z_i | \mathbf{x}_{<t(i)}, \mathbf{z}_{<i}) = \prod_i P(z_i | \mathbf{x}_{<t(i)}, \mathbf{z}_{<i})$ is in general not equal to $P(\mathbf{z} | \mathbf{x})$ anymore. In other words, using a streaming locally normalized model means that the estimated alignment sequence posterior is the product of some locally normalized alignment scores which depend only on partial input $\mathbf{x}_{<t(i)}$, and as a result can no longer accurately represent the true conditional distribution. This will bias the model towards predictions with low-entropy estimated posterior probabilities at each decoding step. This degrades the model ability to revise previous decisions, a phenomenon called *label bias* [24, 2].

### 2.2 Global Normalization

Traditionally, *globally normalized models* such as Conditional Random Fields (CRF) [24] are used to address the label bias problem. This paper seeks to apply global normalization to modern neural architectures that are more similar to CTC, RNN-T, or LAS, rather than traditional linear models, with the purpose of addressing label bias problem for streaming ASR models.

A globally normalized model does not constrain the alignment score $\omega(z_i | \mathbf{x}, \mathbf{z}_{<i})$ to be locally normalized; it only requires it to be any non-negative score, as long as the *denominator* $Z(\mathbf{x}) = \sum_{\mathbf{z}} \prod_i \omega(z_i | \mathbf{x}, \mathbf{z}_{<i})$ is finite. The finite denominator allows us to interpret $\frac{\prod_i \omega(z_i | \mathbf{x}, \mathbf{z}_{<i})}{Z(\mathbf{x})}$ as a conditional probability distribution $P_\omega(\mathbf{z} | \mathbf{x})$. Minimum negative log-conditional-likelihood training can be more expensive for globally normalized models due to the need to compute $Z(\mathbf{x})$ and the corresponding gradients. However with our proposed modular framework, globally normalized model training can be made practical with careful modelling choices on modern hardware.

Any locally normalized model is by definition also globally normalized, whose $Z(\mathbf{x}) = 1$. While the reverse is not true in general, Appendix E shows that under non-streaming settings with a mild condition, locally and globally normalized models actually express the same class of conditional

distributions. Based on this observation, we argue that under non-streaming settings, with adequately powerful neural architectures, maximum log-conditional-likelihood training should yield behaviorly similar locally or globally normalized models, and thus similar WERs in testing. Results from [37, 17] and our own experiments in Section 6 validate this.

## 3  A Modular Framework for Neural ASR

In this section, we introduce a modular framework for neural ASR, using the weighted finite state automaton (WFSA) formalism to calculate the conditional probabilities via alignment scores. The modular framework clearly expresses the modelling choices and enables practical globally normalized model training and inference. We use the WFSA formalism as the language for describing our framework because of its succinctness and precision. Our algorithms are implemented from scratch to run on TPUs, without using existing toolkits such as OpenFst [1], Kaldi [33] or K2 [21].

### 3.1  Preliminaries

We begin with an introduction to the relevant concepts and notations.

A *semiring* $(\mathbb{K}, \oplus, \otimes, \bar{0}, \bar{1})$ consists of a set $\mathbb{K}$ together with an associative and commutative operation $\oplus$ and an associative operation $\otimes$, with respective identities $\bar{0}$ and $\bar{1}$, such that $\otimes$ distributes over $\oplus$, and $\bar{0} \otimes x = x \otimes \bar{0} = \bar{0}$. The *real* semiring $(\mathbb{R}_+, +, \times, 0, 1)$ is used when the weights represent probabilities. The *log* semiring $(\mathbb{R} \cup \{-\infty\}, \oplus_{\log}, +, -\infty, 0)$, isomorphic to the real semiring via the log mapping, is often used in practice for numerical stability.[2] The *tropical* semiring $(\mathbb{R} \cup \{-\infty\}, \max, +, -\infty, 0)$ is often used in shortest-path applications.

A *weighted finite-state automaton* (WFSA) $A = (\Sigma, Q, i, F, \rho, E)$ over a semiring $\mathbb{K}$ is specified by a finite alphabet $\Sigma$, a finite set of states $Q$, an initial state $i \in Q$, a set of final states $F \subseteq Q$, a final state weight assignment $\rho : F \to \mathbb{K}$, and a finite set of transitions $E \subseteq Q \times (\Sigma \cup \{\epsilon\}) \times \mathbb{K} \times Q$ ($\epsilon$ denotes the empty label sequence). Given a transition $e \in E$, $p[e]$ denotes its origin or previous state, $n[e]$ its destination or next state, $o[e]$ its label, and $\omega[e]$ its weight. A *path* $\pi = e_1 \ldots e_k$ is a sequence of consecutive transitions $e_i \in E$: $n[e_{i-1}] = p[e_i]$, $i = 2, \ldots k$. The functions $n$, $p$, and $\omega$ on transitions can be extended to paths by setting: $n[\pi] = n[e_k]$ and $p[\pi] = p[e_1]$ and by defining the weight of a path as the $\otimes$-product of the weights of its constituent transitions: $\omega[\pi] = \omega[e_1] \otimes \cdots \otimes \omega[e_k]$). An *unweighted finite-state automaton* (FSA) $A = (\Sigma, Q, i, F, E)$ is simply a WFSA whose transitions and final states are all weighted by $\bar{1}$.

$\Pi(Q_1, Q_2)$ is the set of all paths from a subset $Q_1 \subseteq Q$ to a subset $Q_2 \subseteq Q$. $\Pi(Q_1, \mathbf{y}, Q_2)$ is the subset of all paths of $\Pi(Q_1, Q_2)$ with label sequence $\mathbf{y} = y_1 \ldots y_U$, $y_i \in \Sigma$. A path in $\Pi(\{i\}, F)$ is said to be accepting or *successful*. The weight associated by WFSA $A$ to any label sequence $\mathbf{y}$ is given by $A(\mathbf{y}) = \bigoplus_{\pi \in \Pi(\{i\}, \mathbf{y}, F)} \omega[\pi] \otimes \rho(n[\pi])$. The *weight* of $A$ is the $\oplus$-sum of weights of all accepting paths $W(A) = \bigoplus_{\pi \in \Pi(\{i\}, F)} \omega[\pi] \otimes \rho(n[\pi])$. For a semiring $\mathbb{K}$ where $\otimes$ is also commutative, the *intersection* (or Hadamard product) of two WFSA $A_1$ and $A_2$ is defined as: $(A_1 \cap A_2)(\mathbf{y}) = A_1(\mathbf{y}) \otimes A_2(\mathbf{y})$. [28] gives an algorithm to compute the intersection. We can view $\mathbf{y}$ as a WFSA that accepts only $\mathbf{y}$ with weight $\bar{1}$, then $A(\mathbf{y}) = W(A \cap \mathbf{y})$.

### 3.2  Probabilistic Modeling and Inference on Acyclic Recognition Lattices

For any feature sequence $\mathbf{x} = x_1 \ldots x_T$, we use a model with trainable parameters $\theta$ to induce a *recognition lattice* WFSA $A_{\theta, \mathbf{x}} = (\Sigma, Q_{\theta, \mathbf{x}}, i_{\theta, \mathbf{x}}, F_{\theta, \mathbf{x}}, \rho_{\theta, \mathbf{x}}, E_{\theta, \mathbf{x}})$. For a label sequence $\mathbf{y} = y_1 \ldots y_U$, the weight associated with it, $A_{\theta, \mathbf{x}}(\mathbf{y})$, is interpreted under the log semiring as the unnormalized negative log conditional probability $P_\theta(\mathbf{y} \mid \mathbf{x}) = \frac{\exp(A_{\theta, \mathbf{x}}(\mathbf{y}))}{\exp(W(A_{\theta, \mathbf{x}}))} = \frac{\exp(W(A_{\theta, \mathbf{x}} \cap \mathbf{y}))}{\exp(W(A_{\theta, \mathbf{x}}))}$.

The recognition lattice $A_{\theta, \mathbf{x}}$ is designed to be acyclic, and therefore the weight of the automata in both the numerator and denominator above can be efficiently computed by visiting the states of the corresponding WFSA in topological order [27]. See Appendix C for our accelerator friendly version of this algorithm. We can thus train an ASR model by minimizing the negative log-conditional-likelihood on the training corpus $\mathcal{D}$, and choosing $\theta^\star = \arg\min_\theta \mathbb{E}_{(\mathbf{x}, \mathbf{y}) \in \mathcal{D}}[-\log(P_\theta(\mathbf{y} \mid \mathbf{x}))] = \arg\min_\theta \mathbb{E}_{(\mathbf{x}, \mathbf{y}) \in \mathcal{D}}[W(A_{\theta, \mathbf{x}}) - W(A_{\theta, \mathbf{x}} \cap \mathbf{y})]$.

---

[2]$a \oplus_{\log} b = \log(e^a + e^b)$

In general, there can be more than one path in $A_{\theta,\mathbf{x}}$ that accepts the same $\mathbf{y}$. During inference, finding the optimal $\hat{\mathbf{y}} = \arg\max_{\mathbf{y}} P_{\theta}(\mathbf{y}|\mathbf{x})$ requires running the potentially expensive WFSA disambiguation algorithm [30] on $A_{\theta,\mathbf{x}}$. As a cheaper approximation, we instead look for the shortest path $\hat{\pi}$ in $A_{\theta,\mathbf{x}}$ under the tropical semiring, and use the corresponding label sequence as the prediction, again using the standard shortest path algorithm for an acyclic WFSA [27].

## 3.3 Inducing the WFSA

Our framework decomposes the sequence prediction task in ASR into three components, each playing a specific role in inducing the recognition lattice $A_{\theta,\mathbf{x}}$.

- The *context dependency* FSA $C = (\Sigma, Q_C, i_C, F_C, E_C)$ is an $\epsilon$-free, unweighted FSA, whose states encode the history of the label sequence produced so far. $C$ is fixed for a given model, independent of input $\mathbf{x}$.
- The *alignment lattice* FSA $L_T = (\Sigma, Q_T, i_T, F_T, E_T)$ is an acyclic, unweighted FSA, whose states encode the alignment between input frames $\mathbf{x}$ and output labels $\mathbf{y}$. $L_T$ depends on only the length $T$ of input $\mathbf{x}$.
- The *weight* function $\omega_{\theta,\mathbf{x}} : Q_T \times Q_C \times (\Sigma \cup \epsilon) \to \mathbb{K}$. $\omega_{\theta,\mathbf{x}}$ is the only component that contains trainable parameters and requires full access to $\mathbf{x}$. This function defines the transition weights in the recognition lattice $A_{\theta,\mathbf{x}}$.

We will discuss how one can define these components in detail in the next section. With $(C, L_T, \omega_{\theta,\mathbf{x}})$ given, the recognition lattice $A_{\theta,\mathbf{x}}$ is defined as follows:

$$
\begin{aligned}
Q_{\theta,\mathbf{x}} &= Q_T \times Q_C \\
i_{\theta,\mathbf{x}} &= (i_T, i_C) \\
F_{\theta,\mathbf{x}} &= F_T \times F_C \\
E_{A_{\theta,\mathbf{x}}} = \Big\{ & \big((q_a, q_c), y, \omega_{\theta,\mathbf{x}}(q_a, q_c, y), (q_a', q_c')\big) \mid \\
& y \in \Sigma, \ (q_a, y, q_a') \in E_T, \ (q_c, y, q_c') \in E_C \Big\} \\
\cup \Big\{ & \big((q_a, q_c), \epsilon, \omega_{\theta,\mathbf{x}}(q_a, q_c, \epsilon), (q_a', q_c)\big) \mid \\
& (q_a, \epsilon, q_a') \in E_T, \ q_c \in Q_C \Big\} \\
\rho_{A_{\theta,\mathbf{x}}}(q) &= \bar{1}, \ \forall q \in F_{\theta,\mathbf{x}}
\end{aligned}
$$

In other words, the topology (states and unweighted transitions) of the recognition lattice $A_{\theta,\mathbf{x}}$ is the same as the FSA intersection $L_T \cap C$ (see Figure 1c for a concrete example); and the transition weights are defined using $\omega_{\theta,\mathbf{x}}$. The $\epsilon$-freeness of $C$ and the acyclicity of $L_T$ implies that the recognition lattice $A_{\theta,\mathbf{x}}$ is also acyclic.

# 4 Components of a GNAT Model

In this section, we define *globally normalized autoregressive transducer* (GNAT) through the framework above, by specifying each model component.

## 4.1 Context Dependency

GNAT uses an $n$-gram context-dependency defined by $C_n = (\Sigma, Q_n, i_n, F_n, E_n)$, where $Q_n = \Sigma^{\leq n}$ corresponds to a label history of length up to $n$. The initial state $i_n = \epsilon$ is the empty label sequence. The transitions $E_n \subseteq Q_n \times \Sigma \times Q_n$ correspond to truncated concatenation: $E_n = \{(q, y, q') \mid q \in Q, y \in \Sigma\}$ where $q'$ is the suffix of $qy$ with length at most $n$. For example, when $n = 2$, the transition $(ab, c, bc)$ goes from state $ab$ to state $bc$ with label $c$. All states are final: $F_n = Q_n$. See Figure 1a for the FSA $C_1$ when $\Sigma = \{a, b\}$. Appendix C demonstrates how $C_n$ can be efficiently intersected during the shortest distance computation thanks to the highly regular structure of $C_n$.

Although not studied in this paper, our modular framework makes it easy to switch to a more sophisticated context dependency, such as clustered histories often used for context-dependent phone models, or a variable context length as used in $n$-gram language models [29].

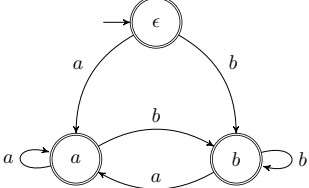

(a) $n$-gram context-dependency automaton $C_1$ for $\Sigma = \{a, b\}$.

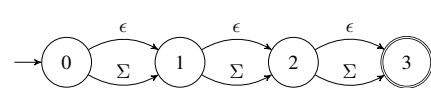

(b) Frame dependent alignment lattice with $T = 3$.

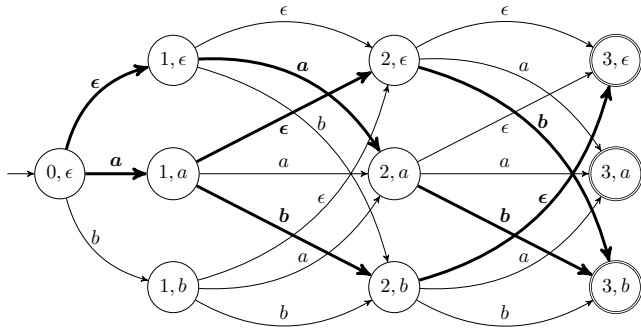

(c) Recognition lattice topology: intersection of (a) and (b) with all paths for output $ab$ highlighted.

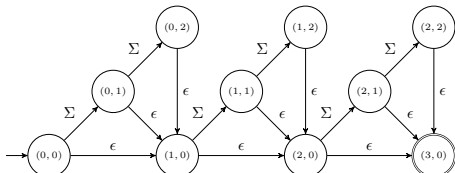

(d) $k$-constrained label and frame dependent alignment lattice with $T = 3, k = 2$.

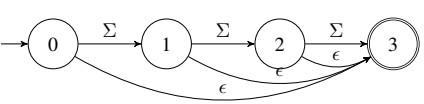

(e) Label dependent alignment lattice with $l(T) = 3$.

Figure 1: Examples of GNAT components

## 4.2 Alignment Lattices

Given the input feature sequence length $T$, the alignment lattice FSA $L_T$ defines all the possible alignments between the feature sequence and allowed label sequences. Since the feature sequence length $T$ usually differs from the label sequence length $U$, many different alignments between the feature sequence and a label sequence can be defined. The states in an alignment lattice FSA encode how the next label or $\epsilon$-transition corresponds to some position in the feature sequence.

We can choose different structures for $L_T$ by encoding one or both of the positions in the feature sequence and the label sequence. A simple example is the *frame dependent alignment* similar to [14], where each frame is aligned to at most one label:

$$Q_T = \{0, \dots, T\}$$
$$i_T = 0$$
$$F_T = \{T\}$$
$$E_T = \big\{(t-1, y, t) \mid y \in \Sigma \cup \{\epsilon\}, \ 1 \le t \le T\big\}$$

Here any state $t < T$ represents a position in the feature sequence. We start by aligning to the initial frame of the feature sequence, and repeatedly shift to the next frame for every subsequent label or $\epsilon$-transition until all frames have been visited. Figure 1b depicts a frame dependent alignment lattice.

To allow a label sequence longer than the feature sequence, we can use the *k-constrained label and frame dependent alignment* similar to [13]:

$$Q_T = \big\{(t, n) \mid 0 \leq t \leq T - 1,\ 0 \leq n \leq k\big\} \cup \big\{(T, 0)\big\}$$

$$i_T = (0, 0)$$

$$F_T = \big\{(T, 0)\big\}$$

$$E_T = \Big\{\big((t, n-1), y, (t, n)\big) \mid$$

$$y \in \Sigma,\ 0 \leq t \leq T - 1,\ 1 \leq n \leq k\Big\}$$

$$\cup \Big\{\big((t-1, k), \epsilon, (t, 0)\big) \mid 1 \leq t \leq T\Big\}$$

Here, up to $k$ consecutive label transitions can align to any single frame. An $\epsilon$-transition is then taken to explicitly shift the alignment to the next frame. The number of labels aligned to one frame is constrained by a constant $k$ solely in order to impose acyclicity. Figure 1d depicts a $k$-constrained label and frame dependent alignment lattice.

Some models may only depend on the position in the label sequence, similar to [8]. In this case we can bound the length of the label sequence by some function $l(T)$, and use the following *label dependent alignment*:

$$Q_T = \big\{0, \ldots, l(T)\big\}$$

$$i_T = 0$$

$$F_T = \{l(T)\}$$

$$E_T = \big\{(u - 1, y, u) \mid y \in \Sigma,\ 1 \leq u \leq l(T)\big\}$$

$$\cup \big\{(u, \epsilon, l(T)) \mid 0 \leq u \leq l(T) - 1\big\}$$

Here, each label can be seen as aligning to the entire feature sequence, and the $\epsilon$-transition serves as an explicit termination of the label sequence. Figure 1e depicts a label dependent alignment lattice.

## 4.3 Weight functions

The weight function $\omega_{\theta, \mathbf{x}}$ translates trainable parameters into transition weights for states in the recognition lattice $A_{\theta, \mathbf{x}}$. The choice of the weight function depends on the choice of the context and alignment lattice FSA, especially the alignment lattice where the meaning of a state directly affects how the weight function can access $\mathbf{x}$.

The experiments discussed in this paper use frame dependent and $k$-constrained label and frame dependent alignment lattices. In these two types of alignment lattices, a non-final state $q_a$ in $Q_T$ contains a position $\tau(q_a)$ in the feature sequence (the state itself in the case of frame dependent alignment lattices; the first value in the state in the case of label and frame dependent alignment lattices). Weight functions can thus be defined in three steps,

1. Feed $\mathbf{x}$ into an encoder, such as unidirectional or bidirectional RNN, or a self-attention encoder to obtain the sequence of hidden units $\mathbf{h}$ of dimension $D$. In the experiments we compared streaming vs non-streaming encoders.
2. Map a single frame of hidden units $\mathbf{h}[t]$ and context state $q_c$ to a $(|\Sigma| + 1)$-dimensional vector, corresponding to the unnormalized transition weights for $y \in \Sigma \cup \{\epsilon\}$.
3. Optionally locally normalize transition weights across $y \in \Sigma \cup \{\epsilon\}$ given $(q_a, q_c)$.

For step 2, we experiment with the following concrete modelling choices with varying degree of parameter sharing,

**Per-state linear projection** (*unshared*) For every context state $q_c$, we obtain a $D \times (|\Sigma| + 1)$ projection matrix $W_{q_c}$ and a $(|\Sigma| + 1)$-dim bias vector $b_{q_c}$ from $\theta$, and define for $y \in \Sigma \cup \{\epsilon\}$:

$$\omega_{\theta, \mathbf{x}}(q_a, q_c, y) = \big(W_{q_c} \cdot \mathbf{h}[\tau(q_a)] + b_{q_c}\big)[y]$$

**Shared linear projection with per-state embedding** (*shared-emb*) We obtain from $\theta$ (a) for every context state $q_c$ a $D$-dimensional state embedding $E_{q_c}$, (b) independent of context states a $D \times (|\Sigma|+1)$ projection matrix $W$ and a $(|\Sigma| + 1)$-dim bias vector $b$, and define for $y \in \Sigma \cup \{\epsilon\}$:

$$\omega_{\theta, \mathbf{x}}(q_a, q_c, y) = \big(W \cdot \tanh(\mathbf{h}[\tau(q_a)] + E_{q_c}) + b\big)[y]$$

**Shared linear projection with RNN state embedding** (*shared-rnn*) Similar to *shared linear projection with per-state embedding* but $E_{q_c}$ is obtained from running an RNN (e.g. LSTM) on the $n$-gram label sequence represented by $q_c$.

## 5  Discussion

**A Modular Framework**    All the existing locally normalized models can be explained within the modular framework presented in Section 3 with a particular choice of context size, alignment lattice and of course with constraining the weights to be locally normalized using softmax function. Appendix B presents how CE, CTC, LAS, RNNT and HAT models can be expressed within this framework. This allows controlled comparison of different components as well as creating new models by mixing different modeling choices. Like locally and globally normalized models in general, when the weights are locally normalized, the denominator of models defined in our framework is one.

Finally, our framework is very different from traditional uses of finite state machines via a cascade of weighted finite state transducer compositions. The separation of the weight function from the automaton topology allows an arbitrarily complex, non-linear weight function to model the dependency among alignment states, context states, and output label, which is impossible with composition cascades. The idea of weight function has been proposed before (e.g. [38]), although as we shall see in experiments, the exact architecture of a weight function plays a crucial role in model accuracy.

**Related Globally Normalized Models**    There is a rich literature on the applications of globally normalized models [5, 26, 35, 24], as well as theoretical studies on the label bias problem [24, 2, 12]. In ASR, there is a lot of research on applying globally normalized models [3, 7, 6, 16, 25, 45, 18]. Among these, MMI [3, 7] is the most relevant to our work. In MMI, a sequence level score is factorized into an acoustic model (AM) score and a language model (LM) score. The denominator is usually approximated over a pruned lattice. Recently [34] introduced lattice-free MMI, which replaces the word level LM with a 4-gram phone LM during training. Both the GNAT and MMI are globally normalized. The GNAT model differs from MMI in several ways: (1) Unlike MMI, GNAT does not require an external LM, nor does it impose any constraint on how the sequence level scores are defined. (2) In MMI training, the LM is kept frozen while the AM parameters are optimized. In GNAT all the model parameters are trained together. (3) GNAT's recognition lattice encodes the same set of weighted alignment paths during training and inference, therefore the denominator computation in GNAT is exactly matched between training and inference. This does not hold for the standard MMI, because by using different LM topologies between training and inference, the denominator computation in training time only serves as an approximation of the denominator during inference.

Implementation-wise, GNAT trains from scratch without any need for initialization or special regularization techniques as used in the lattice-free version of MMI [34]. In addition we were able to train GNAT models with accelerators without any techniques discussed in [34].

The concept of global normalization has also been visited with deep neural networks [17, 10, 42, 44]. These models can be seen as special cases of MMI based on aspects such as how the sequence level scores are factorized, their use of fixed LM for training, and the mismatching training and inference LMs. Thus the comparison between MMI and GNAT applies here as well. Their use of WFST composition cascades results in weaker modelling power compared to our weight function formulation, and their use of non-streaming encoders combined with their model formulation makes globally normalized models equally expressive to the locally normalized counterparts (Appendix E).

**Challenges**    The main challenge with the GNAT model is its scalability to a larger number of label contexts. At each training step, the model requires $O(|Q_C| \times |\Sigma|)$ semiring multiplications and additions. For $n$-gram context dependency, $|Q_c| = (|\Sigma|^{n+1} - 1)/(|\Sigma| - 1)$, thus the computation scale exponentially by value of $n$. However as shown in Appendix D, due to the particular structure of this space the practical computation and memory cost benchmarks do not scale exponentially with $0 \le n \le 2$. We also note that large value of $n$ might also not be necessary: The HAT model [40] reports that a Seq2Seq model with a label history of just the two previous phonemes performs on par with a similar model with a full history trained on very large voice-search corpus. Similar observations are reported in studies with grapheme and wordpiece units [43, 11]. Due to the data sparsity there might not be enough training data to fully represent all $n$-gram context states, so increasing the value of $n$ might not necessarily lead to a WER reduction. One way of dealing with

larger numbers of states is to use standard pruning techniques to keep only some of the most common states in the training data.

## 6  Experiments

**Data**   We use the full 960-hour Librispeech corpus [32] for experiments. The input features are 80-dim. log Mel extracted from a 25 ms window of the speech signal with a 10 ms shift. The SpecAugment library with baseline recipe parameters were used [15]. The ground truth transcription is used without any processing and tokenized by the 28 graphemes that appear in the training data.

**Architecture**   Attention-based architectures allows using the same parameterization for streaming and non-streaming models, thus for all the experiments we used 12-layer Conformer encoders [15] with model dimension 512, followed by a linear layer with output dimension 640. The Conformer parameters are set such that the only difference between streaming and non-streaming models is the right context: at each time frame $t$, the streaming models only access the left context (feature frames from 1 to $t$), while the non-streaming models can see the entire acoustic feature sequence. To enforce the consistency of the encoder architecture between streaming and non-streaming modes, we removed all the sub-architecture which behaved differently between these two modes. Specifically, we removed the convolution sub-sampling layer, and also forced the stacking layers to only stack within the left context. The baseline experiments use a shared-rnn weight function defined in section 4.3. A single layer LSTM is used with 640 cells. The experiments with the unshared weight function use a linear layer of size $(|Q_C| \times |\Sigma|) \times 640$ to project the encoder activation at each time frame into the transition weights of the recognition lattice. In our experiments, $|\Sigma| = 32$. For the $n$-gram context dependency, $|Q_c| = (|\Sigma|^{n+1} - 1)/(|\Sigma| - 1)$. The experiments with the shared-emb weight function use an embedding table of size $|Q_c| \times 128$.

**Training**   All models are trained on $8 \times 8$ TPUs with a batch size 2048. The training examples with more than 1961 feature frames or more than 384 labels are filtered out. We used Adam optimizer [22] ($\beta_1 = 0.9$, $\beta_2 = 0.98$, and $\epsilon = 10^{-9}$) with the Transformer learning rate schedule [41] (10k warm-up steps and peak learning rate $0.05/\sqrt{512}$). We applied the same regularization techniques and the training hyperparameters used in the baseline recipe of [15].

**Evaluation**   We report WER results on standard Librispeech test sets: test_clean and test_other. The WER is either computed with *sum-path* algorithm or *max-path* algorithm. The sum-path algorithm approximately merges the alignment hypothesis corresponding to the same label sequence prefix after removal of epsilons. In ideal decoding, sum-path should result in the most likely output label sequence. The max-path algorithm computes the highest scoring alignment path using algorithms in Appendix C.

**Baselines**   The RNN-T baselines are presented in the rows corresponding to the "*RNN-T*" context dependency in Table 1a. Unlike the other rows in Table 1a, the decoder RNN in RNN-T in principle encodes an unlimited output history as context. The standard RNN-T model effectively uses the label frame dependent alignment lattice with an infinite $k$. The rows corresponding to the frame dependent alignment lattice represent a slightly modified RNN-T model, where each frame is aligned to exactly one output (either a blank, or a lexical label). The WER differences between non-streaming RNN-T baselines in Table 1a and [15] are mainly due to our modifications to the Conformer encoder for a controlled comparison against streaming models.

**Choice of the context dependency**   Table 1a compares effect of $n$-gram context dependency for $n = 1, 2, 3$ and baseline RNN-T models. The general observation is that increasing $n$ leads to better WER independent of the other choices of the modeling parameters. However, the model with 3-gram context dependency already performs on par with RNN-T baseline. The 2-gram context dependency perform almost on par as baseline on clean test set while still lagging on the other test set. This is consistent with the earlier observations in [40, 43, 11].

**Choice of the alignment lattice**   The comparison of different alignment lattices in Table 1a suggests that this choice does not significantly contribute to the model performance. While there is a performance gap for 1-gram context dependency, we do not think there is a principal argument in favor of frame dependent alignment lattice. We speculate that this is more due to the choice of optimization parameters. However, the choice of lattice type can have some side effects. For example as $k$ in $k$-constrained label frame dependent alignment lattice increases, the model has more

| context dep. | alignment lattice | weight fn streaming | WER [%] clean | other |
|---|---|---|---|---|
| 1-gram | frame | no | 4.0 | 10.0 |
| | | yes | 7.1 | 16.0 |
| | label frame | no | 6.7 | 10.2 |
| | | yes | 8.8 | 14.5 |
| 2-gram | frame | no | 2.8 | 6.0 |
| | | yes | 4.9 | 10.0 |
| | label frame | no | 2.5 | 5.6 |
| | | yes | 5.1 | 10.3 |
| 3-gram | frame | no | 2.5 | 5.3 |
| | | yes | 4.9 | 9.7 |
| | label frame | no | 2.5 | 5.3 |
| | | yes | 5.0 | 9.8 |
| RNN-T | frame | no | 2.5 | 5.3 |
| | | yes | 5.1 | 9.8 |
| | label frame | no | 2.5 | 5.5 |
| | | yes | 5.0 | 9.8 |

(a) Locally normalized baselines with different context-dependency and alignment lattice. All models used a shared-rnn weight function with or without a streaming encoder. (sum-path decoding)

| context dep. | weight function streaming | normalization | WER [%] clean | other |
|---|---|---|---|---|
| 1-gram | no | local | 3.4 | 8.7 |
| | | global | 3.3 | 8.4 |
| | yes | local | 7.0 | 17.4 |
| | | global | 5.5 | 14.0 |
| 2-gram | no | local | 2.8 | 6.7 |
| | | global | 2.8 | 6.7 |
| | yes | local | 4.9 | 11.0 |
| | | global | 3.8 | 9.5 |

(b) Weight function parameters: normalization and streaming. (max-path decoding)

| weight function type | normalization | WER [%] clean | other |
|---|---|---|---|
| unshared | local | 4.9 | 10.7 |
| | global | 4.2 | 10.6 |
| shared-emb | local | 5.4 | 13.1 |
| | global | 4.1 | 9.9 |
| shared-rnn | local | 4.9 | 11.0 |
| | global | 3.8 | 9.5 |

(c) Comparison of different weight function types (max-path decoding).

Table 1: Experiment results

ability to delay its prediction to the end of the signal. This implicit lookahead can translate into performance gains particularly for unidirectional models. By limiting this quantity to 1, we observed that performance on clean and other sets degrades by $34.9\%$ and $36.7\%$, respectively.

**Choice of the weight function normalization**   Table 1b examines the effect of weight normalization on non-streaming and streaming models. Here we present models with 1-gram and 2-gram context dependency with a frame dependent alignment lattice. Note that for 0-gram context dependency with a frame dependent alignment lattice it is easy to show that locally normalized and globally normalized models are equivalent. For non-streaming models, the normalization seems to not have an impact on the performance quality neither for 1-gram nor for 2-gram context dependency experiments. This is validates our argument in the end of Section 2.2 that locally and globally normalized models express the same class of conditional distributions under this particular setting. On the other hand, streaming models significantly benefit from global normalization: For the clean test set, the globally normalized model outperforms the locally normalized model by about $21\%$ relative WER for 1-gram context dependency and by about $20\%$ relative gain for 2-gram context dependency.

The globally normalized model with 2-gram context dependency also beat the baseline streaming RNN-T model in Table 1a and performs significantly closer to the non-streaming RNN-T baseline. The equivalent streaming RNN-T model performs $5.1\%$ on test clean and the non-streaming model performs $2.5\%$ on same test set. The globally normalized model decoded with max-path algorithm performs $3.8\%$ on same test set. The globally normalized model effectively closed almost $50\%$ of the performance gap between streaming and non-streaming models.

The reported performance for the globally normalized models is from max-path decoding, while the baselines benefit from sum-path decoding. Comparing the locally normalized models' WER from max-path decoding in Table 1b and their counterparts in Table 1a, it is clear that sum-path decoding leads to an extra WER gain. This gain is more significant on test_other. So we expect the globally model performs even better when decoded with sum-path.

The standard sum-path algorithms use several heuristics particularly for path merging and pruning. While similar merging techniques can be applied to the globally normalized models, the pruning heuristics require several adjustments. This is particularly due to the nature of the globally normalized models where the transition weights are not constrained and can take any value, unlike locally normalized models where the transition weights are constrained to be positive number between 0 and 1 and sum to 1 for all the weights leaving the same state in recognition lattice. We plan to explore approaches to improve globally normalized model decoding in the future.

**Choice of the weight function architecture** Finally Table 1c compares different choices of the architectures for a streaming model with 2-gram context dependency and frame dependent alignment lattice. While the unshared and shared-rnn architectures are very different in terms of parameter sharing among states, both perform well, though the shared-rnn architecture performs slightly better. The shared-emb architecture performs significantly worst than shared-rnn architecture. Note that the shared-rnn model is able to learn common structures across states in the context dependency while shared-emb does not have such capability.

## 7 Conclusion

The GNAT model was proposed and evaluated with the focus on the label bias problem and its impact on the performance gap between streaming and non-streaming locally normalized ASR. The finite context property of this model allows exact computation of the sequence level normalization which makes this model differ from existing globally normalized models. Furthermore, the same property allows accelerator friendly training and inference. We showed that the streaming models with globally normalized criteria can significantly close the gap between streaming and non-streaming models by more than 50%. Finally, the modular framework introduced in this paper to explain the GNAT model encompasses all the common neural speech recognition models. This enables fair and accurate comparison of different models via controlled modelling choices and creation of new ASR models.

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
