# A  An overview example

To illustrate different components of the GNAT model, here we present a toy example of designing a speech recognition for finite alphabet $\Sigma = \{a, b\}$. Given an input feature sequence $\mathbf{x} = x_1 \ldots x_T$, we wish to predict the corresponding output label sequence $\mathbf{y} = y_1 \ldots y_U$, $y_i \in \Sigma$. Our objective is to create a conditional probabilistic model $P(\mathbf{y}|\mathbf{x})$ which assigns the highest probability to the correct label sequences for any given feature sequence. We construct a GNAT model with the following components:

- Context dependency: 2-gram
- Alignment lattice: frame dependent
- Weight function: per-state linear projection, streaming

Next we elaborate details of each of these modules and how they are integrated to create the final space the recognition lattice $A_{\theta,\mathbf{x}}$ and the probabilistic model $P(\mathbf{y}|\mathbf{x})$ as described in Section 3.

## A.1  Context Dependency FSA

Figure 2 presents the 2-gram context dependency $C_2$. The set of states for this space are the initial state, 1-gram states and 2-gram states:

$$Q_C = \{\epsilon, a, b, aa, ab, ba, bb\}$$

With the lexicographic order, these states are indexed as follows:

| state | state index |
|:-----:|:-----------:|
| $\epsilon$ | 0 |
| $a$ | 1 |
| $b$ | 2 |
| $aa$ | 3 |
| $ab$ | 4 |
| $ba$ | 5 |
| $bb$ | 6 |

For this particular FSA, the transitions space is

$$E_C = \big\{ (q, y, q') \,|\, q \in Q, y \in \Sigma) \big\}$$

where $q'$ is the suffix of $qy$ with length at most 2. All 14 transitions of this space are listed in the following table:

| from state | label | to state |
|:----------:|:-----:|:--------:|
| $\epsilon$ | $a$ | $a$ |
| $\epsilon$ | $b$ | $b$ |
| $a$ | $a$ | $aa$ |
| $a$ | $b$ | $ab$ |
| $b$ | $a$ | $ba$ |
| $b$ | $b$ | $bb$ |
| $aa$ | $a$ | $aa$ |
| $aa$ | $b$ | $ab$ |
| $ab$ | $a$ | $ba$ |
| $ab$ | $b$ | $bb$ |
| $ba$ | $a$ | $aa$ |
| $ba$ | $b$ | $bb$ |
| $bb$ | $a$ | $ba$ |
| $bb$ | $b$ | $bb$ |

## A.2  Alignment Lattice FSA

Figure 3 depicts a frame dependent alignment lattice $L_4$ for four frames feature sequence $\mathbf{x}$. The states of this space are:

$$Q_T = \{0, 1, 2, 3, 4\}$$

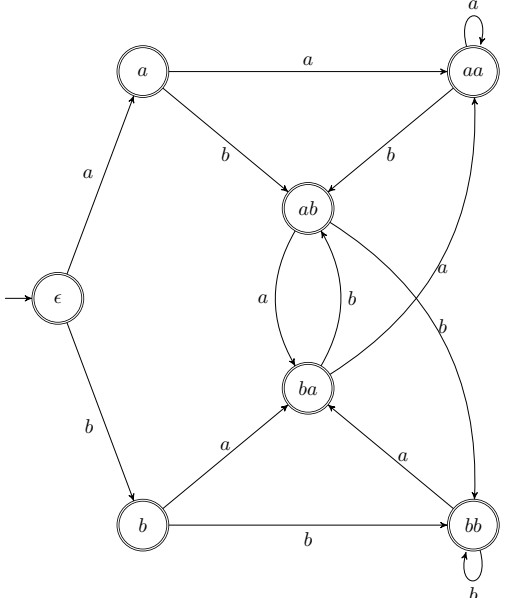

Figure 2: 2-gram context-dependency automaton $C_2$ for $\Sigma = \{a, b\}$.

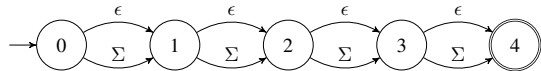

Figure 3: Frame dependent alignment lattice with $T = 4$.

where $0$ is the initial state and $4$ is the final state. Every path starting from the initial state in this automaton corresponds to one possible alignment sequence of the input feature sequence. The example FSA in Figure 1b encodes $3^4 = 81$ possible alignment sequences. An alignment path $\epsilon a \epsilon b$ corresponds to the following sequence of transitions in $L_4$:

$$(0, \epsilon, 1), (1, a, 2), (2, \epsilon, 3), (3, b, 4)$$

### A.3 Weight Function

The weight function $\omega_{\theta,\mathbf{x}} : Q_T \times Q_C \times (\Sigma \cup \epsilon) \to \mathbb{K}$. $\omega_{\theta,\mathbf{x}}$ is the only trainable component of the GNAT model which assigns a weight to every transition of the recognition lattice $A_{\theta,\mathbf{x}}$. We first feed $\mathbf{x} = x_1, ..., x_4$ into an encoder to obtain hidden activations $\mathbf{h} = h_1, ..., h_4$ of dimension $D$. The encoder can be any neural architecture such as DNNs, CNNs, RNNs or Transformers. Since we are interested in streaming weight function for this example, we need to make sure the encoder is also streaming. This means $h_t$ can only depends on $x_{1:t}$. Finally we define a $D \times 3$ matrix $W_{q_c}$ and a 3-dim bias vector $b_{q_c}$ for any $q_c \in Q_c$. The weight function is then defined as[3]:

$$\omega_{\theta,x_{1:t}}(q_a = t, q_c, y) = \exp\left(W_{q_c}[y, :] \cdot h_t + b_{q_c}[y]\right)$$

where $W_{q_c}[y, :]$ is the row of $W_{q_c}$ corresponding to label $y \in \Sigma \cup \{\epsilon\}$. The total number of trainable parameters is $7 \times D \times 3 + 7 \times 3$ parameters plus the number of parameters of the encoder function.

### A.4 The recognition lattice $A_{\theta,\mathbf{x}}$

Given the context dependency FSA $C_2$, the frame dependent alignment lattice $L_4$ and the weight function $\omega_{\theta,\mathbf{x}}$, we are ready to derive the recognition lattice $A_{\theta,\mathbf{x}}$. The state space has $5 \times 7$ states:

$$Q_{\theta,\mathbf{x}} = \left\{(t, q_c) \mid 0 \le t \le 4, q_c \in Q_c\right\}$$

---

[3]Note that for simplicity we use real semiring for all the score calculations in this example.

The transitions in this space is specified by the state it is originated from, $(t, q_c)$, the label $y \in \Sigma \cup \{\epsilon\}$, weight $\omega_{\theta, \mathbf{x}}(t, q_c, y)$ and the state the transition is ended to $(t+1, q_c')$. For the alignment sequence of our example, $\epsilon a \epsilon b$, the transitions are:

| from | label | weight | to |
|------|-------|--------|-----|
| $(0, \epsilon)$ | $\epsilon$ | $\omega_{\theta, x_1}(q_a = 0, q_c = \epsilon, y = \epsilon) = \exp\left(W_0[0, :] \cdot h_1 + b_0[0]\right)$ | $(1, \epsilon)$ |
| $(1, \epsilon)$ | $a$ | $\omega_{\theta, x_{1:2}}(q_a = 1, q_c = \epsilon, y = a) = \exp\left(W_0[1, :] \cdot h_2 + b_0[1]\right)$ | $(2, a)$ |
| $(2, a)$ | $\epsilon$ | $\omega_{\theta, x_{1:3}}(q_a = 2, q_c = a, y = \epsilon) = \exp\left(W_1[0, :] \cdot h_3 + b_1[0]\right)$ | $(3, a)$ |
| $(3, a)$ | $b$ | $\omega_{\theta, x_{1:4}}(q_a = 3, q_c = a, y = b) = \exp\left(W_1[2, :] \cdot h_4 + b_1[2]\right)$ | $(4, ab)$ |

The product of the above weights is the score that the GNAT model assigns to the features sequence $\mathbf{x}$ and alignment sequence $\epsilon a \epsilon b$:

$$\text{score}(\mathbf{x}, \epsilon a \epsilon b) = \omega_{\theta, x_1}(0, \epsilon, \epsilon) \omega_{\theta, x_{1:2}}(1, \epsilon, a) \omega_{\theta, x_{1:3}}(2, a, \epsilon) \omega_{\theta, x_{1:4}}(3, a, b)$$

The GNAT model formulates the posterior probability $P(\mathbf{y}|\mathbf{x})$ by ratio of two quantities:

- numerator: sum of all the $\text{score}(\mathbf{x}, \mathbf{z})$ where $\mathbf{z}$ is an alignment between $\mathbf{x}$ and $\mathbf{y}$. For example if $\mathbf{y} = ab$, there are only 6 possible alignments: $ab\epsilon\epsilon$, $a\epsilon b\epsilon$, $a\epsilon\epsilon b$, $\epsilon ab\epsilon$, $\epsilon a\epsilon b$, $\epsilon\epsilon ab$

- denominator: sum of all the $\text{score}(\mathbf{x}, \mathbf{z})$ where $\mathbf{z}$ can be any $(|\Sigma| + 1)^T = 3^4 = 81$ sequences.

Since the numerator computation is a special case of the denominator, we only present the denominator calculation. To follow the computation presented in Section C, we first define forward variable $\alpha_t$ which is a 7-dim real-valued vector where $\alpha_t[j]$ is the total alignment scores reaching to state index $j$ (corresponding to the state indices in $Q_c$) at time $t$:

$$
\begin{array}{ccccccc}
(t, i) & (t, a) & (t, b) & (t, aa), & (t, ab) & (t, ba) & (t, bb) \\
\alpha_t = (\alpha_t[0] & \alpha_t[1] & \alpha_t[2] & \alpha_t[3] & \alpha_t[4] & \alpha_t[5] & \alpha_t[6])
\end{array}
$$

the initial state $\alpha_0$ is a 1-hot vector with $\alpha_0[j] = 1$ iff $j = 0$, the initial state. At every time frame $t$, the transition weight matrix $\Omega_{t, \Sigma}$ is defined for all the transitions in $E_{A_{\theta, \mathbf{x}}}$ where label is an element of $\Sigma$. This matrix is a structured matrix with only $|\Sigma| = 2$ non-zero elements per row:

$$
\Omega_{t, \Sigma} = 
\begin{array}{c}
\\
(t-1, \epsilon) \\
(t-1, a) \\
(t-1, b) \\
(t-1, aa) \\
(t-1, ab) \\
(t-1, ba) \\
(t-1, bb)
\end{array}
\begin{array}{c}
\begin{array}{ccccccc}
(t, \epsilon) & (t, a) & (t, b) & (t, aa), & (t, ab) & (t, ba) & (t, bb)
\end{array} \\
\left(
\begin{array}{ccccccc}
0 & \omega(t, \epsilon, a) & \omega(t, \epsilon, b) & 0 & 0 & 0 & 0 \\
0 & 0 & 0 & \omega(t, a, a) & \omega(t, a, b) & 0 & 0 \\
0 & 0 & 0 & 0 & 0 & \omega(t, b, a) & \omega(t, b, b) \\
0 & 0 & 0 & \omega(t, aa, a) & \omega(t, aa, b) & 0 & 0 \\
0 & 0 & 0 & 0 & 0 & \omega(t, ab, a) & \omega(t, ab, b) \\
0 & 0 & 0 & \omega(t, ba, a) & \omega(t, ba, b) & 0 & 0 \\
0 & 0 & 0 & 0 & 0 & \omega(t, bb, a) & \omega(t, bb, b)
\end{array}
\right)
\end{array}
$$

Similarly we denote $\Omega_{t, \epsilon}$ to be the transition weight matrix for all the transitions in $E_{A_{\theta, \mathbf{x}}}$ where label is $\epsilon$. This matrix is a diagonal matrix corresponding to the weights of the self loops:

$$
\Omega_{t, \epsilon} = 
\begin{array}{c}
\\
(t-1, \epsilon) \\
(t-1, a) \\
(t-1, b) \\
(t-1, aa) \\
(t-1, ab) \\
(t-1, ba) \\
(t-1, bb)
\end{array}
\begin{array}{c}
\begin{array}{ccccccc}
(t, \epsilon) & (t, a) & (t, b) & (t, aa), & (t, ab) & (t, ba) & (t, bb)
\end{array} \\
\left(
\begin{array}{ccccccc}
\omega(t, \epsilon, \epsilon) & 0 & 0 & 0 & 0 & 0 & 0 \\
0 & \omega(t, a, \epsilon) & 0 & 0 & 0 & 0 & 0 \\
0 & 0 & \omega(t, b, \epsilon) & 0 & 0 & 0 & 0 \\
0 & 0 & 0 & \omega(t, aa, \epsilon) & 0 & 0 & 0 \\
0 & 0 & 0 & 0 & \omega(t, ab, \epsilon) & 0 & 0 \\
0 & 0 & 0 & 0 & 0 & \omega(t, ba, \epsilon) & 0 \\
0 & 0 & 0 & 0 & 0 & 0 & \omega(t, bb, \epsilon)
\end{array}
\right)
\end{array}
$$

For our model, the forward variable $\alpha_t$ can be calculated given $\alpha_{t-1}$ and the above weights matrices as:

$$\alpha_t = \alpha_{t-1}'(\Omega_{t, \Sigma} + \Omega_{t, \epsilon})$$

since every transition at time $t + 1$ is either an $\epsilon$ transition or a non-$\epsilon$ transition. Here $\alpha_{t-1}'$ is the transpose of forward variable $\alpha_{t-1}$.

Given the above iterative equation, the forward variable at time $4$ is equal to:

$$\alpha_4 = \alpha_0(\Omega_{1,\Sigma} + \Omega_{1,\epsilon})(\Omega_{2,\Sigma} + \Omega_{2,\epsilon})(\Omega_{3,\Sigma} + \Omega_{3,\epsilon})(\Omega_{4,\Sigma} + \Omega_{4,\epsilon})$$

and the denominator of the GNAT model is equal to $\sum_{j=0}^{6} \alpha_4[j]$. Replacing the real semiring with the tropical semiring in above calculation will allow us to find the most likely alignment sequence.

# B   A Modular Framework

In this section, we demonstrates how the existing and common neural speech recognition models can be expressed within our proposed framework. These models can be divided into two categories: models with zero-context recognition lattices and models with infinite context recognition lattices.

## B.1   Zero-context Recognition lattice

### B.1.1   Cross-Entropy with Alignments

The conventional cross-entropy models with feed-forward neural architectures [31] define the conditional probability of label sequence $\mathbf{y}$ given feature sequence $\mathbf{x}$ by:

$$P_\theta(\mathbf{y}|\mathbf{x}) = \prod_{t=1}^{T} P_\theta(y_t|x_t)$$

where probability factors $P_\theta(y_t|x_t)$ are derived by some neural architecture parameterized by $\theta$:

$$P_\theta(y = y_t|x_t) = \frac{\exp(W[y_t, :] \cdot h_t + b[y_t])}{\sum_{y \in \Sigma} \exp(W[y, :] \cdot h_t + b[y])}$$

where $h_t$ is the encoder activation of dimension $D$ at time frame $t$, $W$ is a weight matrix of shape $|\Sigma| \times D$ and $b$ is a $|\Sigma|$-dim bias vector.

The equivalent GNAT model is configured as follow:

- Context dependency: 0-gram $C_0$
- Alignment lattice: frame dependent without $\epsilon$ transitions
- Weight function (here $\epsilon$ is the initial and only state of $C_0$):
  - $\omega_{\theta,\mathbf{x}}(q_a = t, q_c = \epsilon, y = y_t) \triangleq P_\theta(y = y_t|x_t)$
  - Locally normalized
  - Streaming

The more advanced cross-entropy models use recurrent architectures or transformers as encoder [39]. The only difference between the GNAT equivalent of these models and above configuration is that whether the encoder is streaming or not.

### B.1.2   Supporting CTC Style Label Deduplication

The standard CTC model [14] is very similar to a GNAT model using a frame dependent alignment lattice and a 0-gram context dependency. One key difference is that CTC introduces a deduplication process when turning its model output sequence to a label sequence. Each model output of a CTC model is either a lexical label from $\Sigma$, or the special blank ($\epsilon$) label. To obtain the label sequence, two steps are applied on the model output in order,

1. Maximal consecutive repeated non-blank labels are merged into one (e.g. turning $abbc$ into $abc$, or $abb\epsilon b$ into $ab\epsilon b$);
2. All the $\epsilon$ labels are removed.

As a comparison, paths on the alignment lattice of the GNAT models in the main paper is equivalent to the model outputs in CTC, whereas the $\epsilon$-free label sequence seen by the context dependency is equivalent to the label sequence in CTC. To support the deduplication of repeated non-blank labels, we need to introduce a finite state transducer into our series of finite state machine compositions. Similar to a finite state automaton, a *weighted finite-state transducer* (WFST) $T = (\Sigma, Q, i, F, \rho, E)$ over a semiring $\mathbb{K}$ is specified by a finite alphabet $\Sigma$, a finite set of states $Q$, an initial state $i \in Q$, a

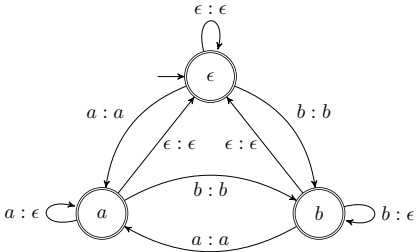

Figure 4: An unweighted FST for CTC style label deduplication with $\Sigma = \{a, b\}$.

set of final states $F \subseteq Q$, a final state weight assignment $\rho : F \to \mathbb{K}$, and a finite set of transitions $E$.[4] The meaning of $\Sigma$, $Q$, $i$, $F$, and $\rho$ are identical to those of a WFSA. The set of transitions $E$ is instead a subset of $Q \times (\Sigma \cup \{\epsilon\}) \times (\Sigma \cup \{\epsilon\}) \times \mathbb{K} \times Q$, i.e. containing a pair of input/output labels instead of just one. A WFSA can be viewed as a WFST with identical input/output labels on each arc, and similar to WFSA intersection, a series of WFST can be composed into a single WFST. We refer the readers to [28] for a full description of WFST and the composition algorithm. Figure 4 is an example unweighted FST, when composed with another input FSA or FST, performs the CTC style label deduplication. More generally, the unweighted label deduplication transducer $D$ of vocabulary $\Sigma$ consists of

- States $Q_D = \Sigma \cup \{\epsilon\}$, $i_D = \epsilon$, $F_D = Q_D$
- Transitions $E = \{(p, x, x, x) | p, x \in Q_D\} \cup \{(x, x, \epsilon, x) | x \in \Sigma\}$

Given a context dependency FSA $C$, an alignment lattice FSA $L_T$, and the weight function $\omega_{\theta,\mathbf{x}}$, as defined in Section 3.3, a GNAT model with CTC style label deduplication induces a WFST $T_{\theta,\mathbf{x}}$ as follows,

$$
\begin{aligned}
Q_{\theta,\mathbf{x}} &= Q_T \times Q_D \times Q_C \\
i_{\theta,\mathbf{x}} &= (i_T, i_D, i_C) \\
F_{\theta,\mathbf{x}} &= F_T \times F_D \times F_C \\
E_{T_{\theta,\mathbf{x}}} &= \Big\{ \big((q_a, q_d, q_c), y, y, \omega_{\theta,\mathbf{x}}(q_a, q_c, y), (q'_a, q'_d, q'_c)\big) \mid \\
&\quad\quad y \in \Sigma, \ (q_a, y, q'_a) \in E_T, \ (q_d, y, y, q'_d) \in E_D, \ (q_c, y, q'_c) \in E_C \Big\} \\
&\quad \cup \Big\{ \big((q_a, q_d, q_c), \epsilon, \epsilon, \omega_{\theta,\mathbf{x}}(q_a, q_c, \epsilon), (q'_a, q'_d, q_c)\big) \mid \\
&\quad\quad (q_a, \epsilon, q'_a) \in E_T, \ (q_d, \epsilon, \epsilon, q'_d) \in E_D, \ q_c \in Q_C \Big\} \\
&\quad \cup \Big\{ \big((q_a, q_d, q_c), y, \epsilon, \omega_{\theta,\mathbf{x}}(q_a, q_c, y), (q'_a, q'_d, q_c)\big) \mid \\
&\quad\quad y \in \Sigma, \ (q_a, y, q'_a) \in E_T, \ (q_d, y, \epsilon, q'_d) \in E_D \Big\} \\
\rho_{T_{\theta,\mathbf{x}}}(q) &= \bar{1}, \ \forall q \in F_{\theta,\mathbf{x}}
\end{aligned}
$$

In other words, the topology of $T_{\theta,\mathbf{x}}$ is the same as the following cascade of FST compositions,

1. $D \cdot C$ treating output $\epsilon$ labels in $D$ as empty (i.e. standard FST composition).
2. $L_T \cdot (D \cdot C)$ treating $\epsilon$ transitions in $L_T$ and input $\epsilon$ labels in $(D \cdot C)$ as regular labels.

and the transition weights are defined using $\omega_{\theta,\mathbf{x}}$ just like the GNAT models in the main paper.

When implemented naively, CTC style label deduplication causes a $|Q_D| = (|\Sigma| + 1)$ blow up in $|Q_{\theta,\mathbf{x}}|$. However, by inferring about states in $Q_D$ from states in $Q_C$, we can greatly reduce the number of states needed. For each state $x \in \Sigma$ in $Q_D$, we know the last non-blank label observed when reaching state $x$ must be label $x$. Similarly, for context dependencies we care about ($n$-gram and string), there is a unique label $x(q_c)$ for all incoming arcs of each non-start state $q_c$ (start states do

---

[4]Here we make the simplification that the input and output vocabularies are identical, i.e. $\Sigma$.

not have any incoming arcs in these context dependencies). Thus, the states in $Q_{\theta,\mathbf{x}}$ that are reachable from the start must match one of the following patterns,

- $(q_a, \epsilon, q_c)$, $\forall q_a \in Q_A, q_c \in Q_X$
- $(q_a, x(q_c), q_c)$, $\forall q_a \in Q_A, q_c \in Q_X \setminus \{i_X\}$

This means the actual number of states we shall visit in computing the shortest distance is only $2|Q_A||Q_C|$.

## B.2 Infinite Recognition Lattices with Finite Intersection

To describe models such as *Listen, Attend and Spell* and *Recurrent Neural Transducer* in our framework, we need to allow the complete recognition lattice to be infinite, because of the possibly infinite context dependency or alignment lattices admitted by such models.

Despite the infinite size of the context dependency and alignment lattices, implying an infinite size of the recognition lattice, the recognition lattice intersected with a reference string still produces a finite state machine. This is why locally normalized training is feasible for such models. However, their globally normalized counterparts are in general not well defined because the denominator is usually unbounded.

### B.2.1 Listen, Attend and Spell (LAS)

This model formulates the posterior probability by directly applying chain rule (Equation 1 in [8]):

$$P_\theta(\mathbf{y}|\mathbf{x}) = \prod_l P_\theta(y_l|\mathbf{x}, \mathbf{y}_{<l})$$

the posterior factors are defined as (Equations 6 to 8 of [8]):

$$P_\theta(y_l|\mathbf{x}, \mathbf{y}_{<l}) = \text{CharacterDistribution}(s_l, c_l)$$

where

$$
\begin{aligned}
\mathbf{h} &= \text{Listen}(\mathbf{x}) \\
s_l &= \text{RNN}(s_{l-1}, y_{l-1}, c_{l-1}) \\
c_l &= \text{AttentionContext}(s_l, \mathbf{h})
\end{aligned}
$$

here Listen is a bidirectional encoder function, AttentionContext is the attention network (Equations 9 to 11 of [8]).

The equivalent GNAT model is configured as follow:

- Context dependency: $\infty$-gram
- Alignment lattice: label dependent since the probability factorizes only on label sequence.
- Weight function:
  - $\omega_{\theta,\mathbf{x}}(q_a = l, q_c = q, y = y_l) \triangleq P_\theta(y = y_l|\mathbf{x}, \mathbf{y}_{<l})$
  - Locally normalized
  - Non-streaming

### B.2.2 Recurrent Neural Transducer

The RNNT model formulate the posterior probability as marginalization of alignment sequences (Equation 1 in the RNNT paper [13]):

$$P_\theta(\mathbf{y}|\mathbf{x}) = \sum_{\mathbf{z} \in B^{-1}(\mathbf{y})} P_\theta(\mathbf{z}|\mathbf{x})$$

where $\mathbf{z} = z_1, \cdots, z_{T+L}$ is an alignment sequence, $z_i \in \Sigma \cup \{\epsilon\}$, $T$ is the number of acoustic frames and $L$ is the number of labels. The function $B(\mathbf{z}) = \mathbf{y}$ removes the epsilons from the alignment sequence. The alignment posterior is factorized along the alignment path as:

$$P_\theta(\mathbf{z}|\mathbf{x}) = \prod_{j=1}^{T+L} P_\theta(z_j|\mathbf{x}, \mathbf{z}_{<j})$$

and finally RNNT make the following assumption:

$$P_\theta(z_j|\mathbf{x}, \mathbf{z}_{<j}) = P_\theta(z_j|\mathbf{x}, B(\mathbf{z}_{<j}) = \mathbf{y}_{<u})$$

which means if the prefix of two alignments be equal after epsilon removal, the model assigns same expansion probability for the next alignment position. The inner terms in the above equation is defined (Equations 12 to 15 of the RNNT paper):

$$P_\theta(z_j|\mathbf{x}, B(\mathbf{z}_{<j}) = \mathbf{y}_{<u}) = \frac{\exp(W[z_j,:] \cdot (h_{j-u} + g_u) + b[z_j])}{\sum_{y \in \Sigma \cup \{\epsilon\}} \exp(W[y,:] \cdot (h_{j-u} + g_u) + b[y])}$$

where $h_{j-u}$ is the encoder activation at time frame $j - u$ (referred to as the transcription network in [13]) and $g_u$ is the output of the prediction network which is a simple stack of RNNs.

The equivalent GNAT model is configured as follow:

- Context dependency: $\infty$-gram
- Alignment lattice: $\infty$-constrained label and frame dependent
- Weight function:
  - $\omega_{\theta,\mathbf{x}}(q_a = (t,u), q_c = q, y = z_{t+u+1}) \triangleq P_\theta(z_{t+u+1}|\mathbf{x}, B(\mathbf{z}_{<t+u+1}) = \mathbf{y}_{<u})$
  - locally normalized
  - non-streaming

While the original definition of the RNNT model is based on non-streaming encoder (transcription network), this model is widely used for streaming applications by using a streaming encoder. This is in contradiction of the forward-backward derivations in the original paper which explicitly assumes dependency on the whole sequence for any position of alignment sequence (Equation 17 of [13])

Similar to RNNT, the hybrid autoregressive transducer (HAT) [40] model can be also configured in the GNAT framework with the same parametrization as RNNT. The only difference is the weight function. The HAT model defines different probabilities for label transitions and $\epsilon$-transitions (duration model in [40]):

$$P_\theta(z_j|\mathbf{x}, B(\mathbf{z}_{<j}) = \mathbf{y}_{<u}) = \begin{cases} b_{j-u,u} & z_j = \epsilon \\ (1 - b_{j-u,u}) P_\theta(y_{u+1}|X, B(\mathbf{z}_{<j}) = \mathbf{y}_{<u}) & z_j \in \Sigma \end{cases} \tag{1}$$

where $b_{t,u}$ is a sigmoid function defined in Equation 6 of [40].

## C  Accelerator-Friendly Computation

The standard shortest distance/path algorithm for acyclic WFSA [27] can be used for training (computing $W(A)$ for some acyclic $A$) and inference of a GNAT model. To compute $W(A)$ for an acyclic WFSA $A$, we maintain the following forward weight $\alpha_q$ for each state $q$ in $Q_A$:

$$\alpha_q = \begin{cases} \bar{1} & \text{if } q = i_A, \\ \bigoplus_{(p,y,w,q) \in E_A} \alpha_p \otimes w & \text{else.} \end{cases}$$

The weight of $A$ is then $W(A) = \bigoplus_{q \in F_A} \alpha_q \otimes \rho_A(q)$. The recurrence in the definition of $\alpha_q$ can be computed by visiting states in $Q_A$ in a topological order.

To make better use of the compute power of modern accelerator hardware, we observe the following properties of the $C$ or $L_T$ presented so far that enable us to use a more vectorized variant of the shortest distance algorithm in Figure 5:

- From any topological ordering on $Q_T$, we can derive a topological ordering on $Q_{\theta,\mathbf{x}}$.
- The $n$-gram context dependency FSA $C_n$ is deterministic, namely leaving any state there is no more than 1 transition for any label $y \in \Sigma$, and there is no $\epsilon$-transition.
- For all three types of alignment lattices, for any non-final state $q \in Q_T \setminus F_T$, there is a unique next state $\text{succ}(q)$ for transitions leaving $q$ consuming any label $y \in \Sigma$.

Center to an efficient implementation of the algorithm in Figure 5 is the function $\text{next}_C$. This function receives as input the current forward weight vector $\bar{\alpha}_{q_a}$ for states $(q_a, q_c)$, $\forall q_c \in Q_C$, and the transition weights for leaving these states via label transitions, and returns the forward weights

{Initialize the length $|Q_C|$ forward weight vectors $\bar{\alpha}_{q_a}$}
**for all** $q_a \in Q_T$ **do**
   $\bar{\alpha}_{q_a} \leftarrow [\bar{0}, \ldots, \bar{0}]$
**end for**
$\bar{\alpha}_{i_T}[i_C] \leftarrow \bar{1}$
{Compute $\bar{\alpha}_{q_a}$ for $q_a \neq i_T$}
**for all** $q_a \in Q_T$ in topological order **do**
   {$\Omega$ is a $[|Q_C|, |\Sigma| + 1]$ matrix}
   $\Omega \leftarrow \bar{\omega}_{\theta,\mathbf{x}}(q_a, Q_C, \Sigma \cup \{\epsilon\})$
   **if** $q_a$ has outgoing label transitions to $q_a' = \text{succ}(q_a)$ **then**
     $\bar{\alpha}_{q_a'} \leftarrow \bar{\alpha}_{q_a'} \bar{\oplus} \text{next}_C(\bar{\alpha}_{q_a}, \Omega[:, \Sigma])$
   **end if**
   **for all** $q_a'$ such that $(q_a, \epsilon, q_a') \in E_T$ **do**
     $\bar{\alpha}_{q_a'} \leftarrow \bar{\alpha}_{q_a'} \bar{\oplus} (\bar{\alpha}_{q_a} \bar{\otimes} \Omega[:, \epsilon])$
   **end for**
**end for**
**return** $\bigoplus_{q_a \in F_T, q_c \in F_C} \bar{\alpha}_{q_a}[q_c]$

Figure 5: The vectorized shortest distance algorithm for $A_{\theta,\mathbf{x}}$. We denote $\bar{\oplus}$, $\bar{\otimes}$, and $\bar{\omega}_{\theta,\mathbf{x}}$ the vectorized versions of the corresponding operations.

{Inputs: $\bar{\alpha}_{q_a}$ and $\Omega[:, \Sigma]$}
**if** $n = 1$ **then**
   {$Q_{C_n}$ contains only $i_{C_n}$}
   **return** $\bar{\alpha}_{q_a} \bar{\otimes} \bigoplus_{y \in \Sigma} \Omega[i_{C_n}, y]$
**end if**
{Initialize length $|Q_{C_n}|$ vector $\bar{\alpha}$}
$\bar{\alpha} \leftarrow [\bar{0}, \ldots, \bar{0}]$
{States in $Q_{C_n}$ are numbered from 0 to $|Q_{C_n}| - 1 = \sum_{i=1}^{n-1} |\Sigma|^i$ following the lexicographic order}
$l \leftarrow 0$
**for** $i = 0$ **to** $n - 2$ **do**
   $h \leftarrow l + |\Sigma|^i$
   $\bar{\alpha}[l \cdot |\Sigma| + 1 : h \cdot |\Sigma| + 1] \leftarrow \text{flatten}(\Omega[l : h, \Sigma])$
   $l \leftarrow h$
**end for**
**for** $i = 0$ **to** $|\Sigma| - 1$ **do**
   $\bar{\alpha}[l :] \leftarrow \bar{\alpha}[l :] \bar{\oplus} \text{flatten}(\Omega[l + i \cdot |\Sigma|^{n-2} : l + (i+1) \cdot |\Sigma|^{n-2}, \Sigma])$
**end for**
**return** $\bar{\alpha}$

Figure 6: Specialized implementation of $\text{next}_{C_n}$. The flatten function flattens a matrix into a vector by joining the rows.

going to states $(q_a', q_c')$ by taking the $(q_c, y, q_c')$ transitions for $y \in \Sigma$. In other words, $\text{next}_C[q_c'] = \bigoplus_{(q_c, y, q_c') \in E_C} \bar{\alpha}_{q_a}[q_c] \otimes \Omega[q_c, y]$. The $n$-gram context dependency $C_n$ allows a particularly simple and efficient implementation of $\text{next}_{C_n}$, as outlined in Figure 6. The key observation is that when we number the states in $Q_C$ following the lexicographic order, the $|\Sigma|$ transitions leaving the same $q_c$ lead to states in a consecutive range $[\sigma(q_c y_0), \ldots, \sigma(q_c y_{|\Sigma|-1})]$, where $[y_0, \ldots, y_{|\Sigma|-1}]$ are the lexicographically sorted labels of $\Sigma$, and $\sigma(s)$ is the suffix of label sequence $s$ of length up to $n - 1$.

During training, we also need to compute the shortest distance $D(A_{\theta,\mathbf{x}} \cap \mathbf{y})$. We note the algorithm in Figure 5 can also be used for this purpose since $(L_T \cap C) \cap \mathbf{y} = L_T \cap (C \cap \mathbf{y})$, and we simply need to substitute $C$ with $C \cap \mathbf{y}$ in the algorithm.

# D   Memory and Computation Time Benchmarks

The memory and computation benchmark of our implementation for the GNAT model is presented in Table 2. We present benchmarks for training and inference for different configurations of the GNAT model:

- Context dependency: 0-gram, 1-gram and 2-gram

- Alignment lattice: frame dependent, 1-constrained label and frame dependent

- Weight functions: Per-state linear projection (unshared), Shared linear projection with per-state embedding (shared-emb), Shared linear projection with RNN state embedding (shared-rnn)

| context dependency | alignment lattice | weight function | | memory [M] | | time [sec] | |
| --- | --- | --- | --- | --- | --- | --- | --- |
| | | type | normalization | train | decode | train | decode |
| 0-gram | frame | unshared | local | 126.47 | 64.97 | 0.15 | 0.02 |
| | | | global | 124.58 | 65.19 | 0.14 | 0.02 |
| | | shared-emb | local | 124.20 | 65.12 | 0.20 | 0.02 |
| | | | global | 124.64 | 65.19 | 0.18 | 0.02 |
| | | shared-rnn | local | 124.43 | 65.16 | 0.20 | 0.02 |
| | | | global | 124.88 | 65.29 | 0.18 | 0.02 |
| | label frame | unshared | local | 174.62 | 65.00 | 0.16 | 0.02 |
| | | | global | 172.21 | 65.20 | 0.18 | 0.04 |
| | | shared-emb | local | 172.32 | 65.15 | 0.20 | 0.02 |
| | | | global | 172.25 | 65.17 | 0.22 | 0.03 |
| | | shared-rnn | local | 172.55 | 65.19 | 0.20 | 0.02 |
| | | | global | 172.49 | 65.27 | 0.22 | 0.04 |
| 1-gram | frame | unshared | local | 144.04 | 64.95 | 0.17 | 0.044 |
| | | | global | 146.23 | 65.51 | 0.19 | 0.05 |
| | | shared-emb | local | 156.02 | 70.42 | 0.22 | 0.05 |
| | | | global | 158.12 | 70.52 | 0.22 | 0.05 |
| | | shared-rnn | local | 157.04 | 70.67 | 0.23 | 0.05 |
| | | | global | 159.15 | 70.76 | 0.23 | 0.05 |
| | label frame | unshared | local | 192.19 | 64.98 | 0.18 | 0.04 |
| | | | global | 192.70 | 65.19 | 0.27 | 0.07 |
| | | shared-emb | local | 204.13 | 70.45 | 0.23 | 0.05 |
| | | | global | 204.65 | 70.14 | 0.29 | 0.07 |
| | | shared-rnn | local | 205.16 | 70.69 | 0.23 | 0.05 |
| | | | global | 205.68 | 70.38 | 0.30 | 0.07 |
| 2-gram | frame | unshared | local | 306.94 | 187.21 | 0.23 | 0.07 |
| | | | global | 513.58 | 195.36 | 1.55 | 0.41 |
| | | shared-emb | local | 156.05 | 70.42 | 0.23 | 0.05 |
| | | | global | 181.42 | 73.49 | 1.16 | 0.23 |
| | | shared-rnn | local | 174.22 | 73.04 | 0.23 | 0.05 |
| | | | global | 199.62 | 76.11 | 1.16 | 0.23 |
| | label frame | unshared | local | 320.98 | 187.24 | 0.24 | 0.07 |
| | | | global | 428.40 | 187.90 | 3.79 | 0.94 |
| | | shared-emb | local | 204.17 | 70.45 | 0.24 | 0.05 |
| | | | global | 210.61 | 71.42 | 2.63 | 0.48 |
| | | shared-rnn | local | 222.33 | 73.07 | 0.24 | 0.05 |
| | | | global | 229.17 | 74.04 | 2.64 | 0.49 |

Table 2:  Memory and computation benchmarks of the GNAT model for different configurations.

For each configuration, the memory usage footprint is presented in terms of MB and total computation time is presented in terms of number of seconds. The benchmarks do not include the memory and computation footprint of the encoder activations. The training benchmarks are corresponding to the calculation of the GNAT criterion as well as all the backward gradient calculation up to the encoder

activations. The evaluation benchmarks only contain the forward pass memory and compute footprint to find the most likely hypothesis.

All the memory and computation benchmarks are evaluated for an input batch of 32 examples each with 1024 number of frames. Each frame is a 512-dim vector corresponding to the encoder activations. Each example in the input batch are assumed to have at most 256 labels. The alphabet size is set to 32.

The main observations are:

- The larger context dependency lead to more memory and compute footprint. This is expected since the computation complexity is directly related to the context dependency state size. However, interestingly, the memory and computation values do not scale exponentially by value of $n$ in $n$-gram context dependency (as a result by number of states in the context dependency).

- Label frame dependent alignment lattice generally leads to higher memory usage and computation time compare to the frame dependent alignment lattice. This is expected since the label frame dependent consist of alignment paths of length $1024 + 256 = 1270$, corresponding to the sum of number of frames and number of labels.

- The per-state linear projection weight function requires more memory and has longer compute time compare to the shared weights function which is expected by design. Both shared weight functions are performing on-par of each other in terms of memory and compute.

- The global normalization requires more memory and time and the difference is more significant for context dependency FSAs with more number of states (2-gram versus 1-gram).

## E Equivalence of Certain Locally and Globally Normalized Models under Non-Streaming Settings

As widely discussed in literature (e.g. Section 4.6.12 of [23], [24, 2]), under streaming settings, a locally normalized model introduces an independence assumption between an output $z_i$ and future inputs $\mathbf{x}_{\geq t(i)}$, when future outputs $\mathbf{z}_{>i}$ are not yet known. On the contrary, a globally normalized model using the same score factorization does not introduce the same independence assumption. This is the key reason to a locally normalized model's inability to express certain conditional distributions and thus prone to the label bias problem.

However, under non-streaming settings, the entire input sequence $\mathbf{x}$ is available for predicting each $z_i$. The aforementioned independence assumption is thus no longer present for locally normalized models. We shall show that under a mild condition regarding how an alignment score $\omega(z_i|\mathbf{z}_{<i}, \mathbf{x})$ makes use of information from the past outputs $\mathbf{z}_{<i}$, for every non-streaming globally normalized model, there is a non-streaming locally normalized model that defines the same conditional distribution for the output sequence.

For an alignment score $\omega(z_i|\mathbf{z}_{<i}, \mathbf{x})$, for each output step $i$, define the *past output index set* $\delta_i \subseteq \{1, 2, \ldots, i-1\}$ to consist of the indices of past outputs on which $\omega(z_i|\mathbf{z}_{<i}, \mathbf{x})$ actually depend. For example, for a GNAT model using a $n$-gram context dependency with $n = 2$, $\delta_i = \{i-1, i-2\}$, i.e. the alignment score only depends on the previous two outputs. On the other hand, an RNN-T model with unlimited history has $\delta_i = \{1, 2, \ldots, i-1\}$. When comparing locally and globally normalized models, it is important to ensure the same past output index sets are used. Otherwise the comparison is not fair since the parameterization of these models will differ.

Given the sequence of past output index sets $[\delta_1, \delta_2, \ldots]$, the alignment score function $\omega(z_i|\mathbf{z}_{<i}, \mathbf{x})$ can also be written as $\omega(z_i|\mathbf{z}_{\delta_i}, \mathbf{x})$, with $\mathbf{z}_{\delta_i} = \{z_j | j \in \delta_i\}$. The sequence of past output index sets *grows monotonically* if for all $i$, $\delta_i \cap \{1, 2, \ldots, i-2\} \subseteq \delta_{i-1}$. In other words, each $\delta_i$ can be constructed from $\delta_{i-1}$ by first removing some elements then optionally adding $i-1$. Importantly if some $j < i$ is not in $\delta_i$, then $j$ is not in any subsequent $\delta_k$ for any $k \geq i$. Notably, all existing work on neural globally normalized ASR models including this paper itself use score functions where the sequence of past output index sets grows monotonically [17, 10, 42, 44].

When the sequence of past output index sets grows monotonically, globally normalized models and locally normalized ones are in fact equivalent in terms of the conditional distributions that they can express. Since any locally normalized model is by definition globally normalized, we only need to show the other direction.

**Theorem E.1.** *For any non-streaming globally normalized model with alignment score $\omega_G(z_i|\mathbf{z}_{\delta_i}, \mathbf{x})$, if the sequence of past index sets $\{\delta_1, \delta_2, \ldots\}$ grows monotonically, then there exists a non-streaming locally normalized model with alignment score of the same sequence of index sets $\omega_L(z_i|\mathbf{z}_{\delta_i}, \mathbf{x})$, such that $P_{\omega_G}(\mathbf{z}|\mathbf{x}) = P_{\omega_L}(\mathbf{z}|\mathbf{x})$.*

This theorem can be seen a result of such a globally normalized model's corresponding Markov network representation being chordal (see Theorem 4.13 of [23]). In an effort to be self-contained , below we instead present a direct proof that does not rely on any results on graphical models, using essentially the same technique as [9, 36].

*Proof.* For any prefix $\mathbf{z}_{\leq i}$, the sum of the scores of all possible suffices continuing $\mathbf{z}_{\leq i}$ is commonly known as the *backward score*. More specifically, in our case, the backward score at position $i$ is

$$\beta_i = \sum_{\mathbf{z}'|\mathbf{z}'_{<i+1}=\mathbf{z}_{<i+1}} \prod_{j=i}^{|\mathbf{z}'|} \omega_G(z'_j|\mathbf{z}'_{\delta_j}, \mathbf{x})$$

It is easy to see that $\beta_i$ only depends $\mathbf{x}$ and $z_j$ for $j \in \{i\} \cup \delta_i$. Thus $\beta_i(z_i, \mathbf{z}_{\delta_i}, \mathbf{x})$ is a well-defined function. $\beta_i$ is upper bounded by the denominator $Z(\mathbf{x})$ and thus finite. $\omega_G(z_j|\mathbf{z}_\delta, \mathbf{x}) > 0$ implies $\beta_i > 0$. We can thus define the following alignment score which is a locally normalized model,

$$\omega_L(z_i|\mathbf{z}_{\delta_i}, \mathbf{x}) = \frac{\beta_i(z_i, \mathbf{z}_{\delta_i}, \mathbf{x})}{\sum_{z'_i} \beta_i(z'_i, \mathbf{z}_{\delta_i}, \mathbf{x})}$$

Then

$$
\begin{aligned}
P_{\omega_G}(z_i|\mathbf{z}_{<i}, \mathbf{x}) &= \frac{\sum_{\mathbf{z}'|\mathbf{z}'_{<i+1}=\mathbf{z}_{<i+1}} \prod_{j=1}^{|\mathbf{z}'|} \omega_G(z'_j|\mathbf{z}'_{\delta_j}, \mathbf{x})}{\sum_{\mathbf{z}'|\mathbf{z}'_{<i}=\mathbf{z}_{<i}} \prod_{j=1}^{|\mathbf{z}'|} \omega_G(z'_j|\mathbf{z}'_{\delta_j}, \mathbf{x})} \\
&= \frac{\left(\prod_{j=1}^{i-1} \omega_G(z_j|\mathbf{z}_{\delta_j}, \mathbf{x})\right) \left(\sum_{\mathbf{z}'|\mathbf{z}'_{<i+1}=\mathbf{z}_{<i+1}} \omega_G(z_i|\mathbf{z}_{\delta_i}, \mathbf{x}) \prod_{j=i+1}^{|\mathbf{z}'|} \omega_G(z'_j|\mathbf{z}'_{\delta_j}, \mathbf{x})\right)}{\left(\prod_{j=1}^{i-1} \omega_G(z_j|\mathbf{z}_{\delta_j}, \mathbf{x})\right) \left(\sum_{\mathbf{z}'|\mathbf{z}'_{<i}=\mathbf{z}_{<i}} \prod_{j=i}^{|\mathbf{z}'|} \omega_G(z'_j|\mathbf{z}'_{\delta_j}, \mathbf{x})\right)} \\
&= \frac{\beta_i(z_i, \mathbf{z}_{\delta_i}, \mathbf{x})}{\sum_{z'_i} \beta_i(z'_i, \mathbf{z}_{\delta_i}, \mathbf{x})} \\
&= \omega_L(z_i|\mathbf{z}_{\delta_i}, \mathbf{x}) \\
&= P_{\omega_L}(z_i|\mathbf{z}_{<i}, \mathbf{x})
\end{aligned}
$$

Therefore, $P_{\omega_G}(\mathbf{z}|\mathbf{x}) = P_{\omega_L}(\mathbf{z}|\mathbf{x})$. $\qquad\qquad\square$