# OpenReview forum: "Global Normalization for Streaming Speech Recognition in a Modular Framework"
_NeurIPS.cc/2022/Conference — NeurIPS 2022 Accept_

### Official Review · Reviewer_tQ4B · 2022-07-08

**Rating:** 7
**Confidence:** 3
**Soundness:** 3 good
**Presentation:** 3 good
**Contribution:** 3 good

**Summary:**

This paper introduces the Globally Normalized Autoregressive Transducer (GNAT) to address the label bias problem in streaming speech recognition.

**Questions:**

see weakness

**Strengths And Weaknesses:**

Strength:
1.This work mitigates the performance gap between the streaming and non-streaming systems by solving the label bias problem. It is a novel idea.

2.Moreover, this work is unlike other works, which simply tune the structure of networks. It is concrete in both theory and implementation. The experimental results in streaming ASR through the proposed global normalization significantly close more than 50% of the WER to a non-streaming system.

3.Theoratical soundness under WFSA framework

weakness:
Here,  the readers would have more expectations of this work.
You still need to compile the WFSA, which requires huge resources and is less flexible without an on-the-fly compiling mechanism.

---

> ### Author Response · Authors · 2022-08-02
> **Response to Reviewer tQ4B**
>
> > weakness: Here, the readers would have more expectations of this work. You still need to compile the WFSA, which requires huge resources and is less flexible without an on-the-fly compiling mechanism.
>
> We appreciate the reviewer’s positive opinion on our paper. Regarding the resource requirements for our algorithms, we would like to note that the FSA operations are implemented from scratch to take advantage of the specific structure of GNAT (the structure matrices shown in Appendix A) and the target TPU accelerator architecture . The recognition lattice WFSA does not have to be materialized in memory for training or inference, so indeed no on-the-fly composition is required. For further details, we refer the reviewer to Appendix C, where we outline our algorithms.

---

### Official Review · Reviewer_wc8G · 2022-07-10

**Rating:** 6
**Confidence:** 5
**Soundness:** 4 excellent
**Presentation:** 3 good
**Contribution:** 2 fair

**Summary:**

The authors tackle in my opinion 2 main subproblems which are relevant:
1. they propose a unified wFST theoretical framework to unify many of the common ASR architectures alignment aware architectures, namely, RNN-T, CTC, HAT, LAS ... and variations thereof as instances of the same theoretical framework. This was partially covered in [16], but not as well developed as in this paper.

2. They show that approximately 50 % of the degradation that happens when going from offline models (Conformer encoders w/o limited attention) to causal encoders (causal convolutions and causal attention) can be reverted back when using globally normalized variants. They also propose an algorithm implementation as pseudo-code in the appendix.

The results are interesting and relevant for the ASR community.

**Questions:**


I think the following points would have added a lot of value to the paper:

* experiments on how the performance changes with the encoder look-ahead
* analysis of different approaches for approximating the denominator sum, e.g. n-best, lattices ,,,
* analysis/experiments on the conditions for which the label bias is  a major problem by for instance what would happen on different datasets or what would happen if we have noisy training data , ... etc,
* the paper uses a very small vocabulary of 30 characters, should the same conclusions arise with word-level models composed of thousands of words/word-pieces ?

Finally, despite the novelty of the globally normalized model, the paper seems related to K2 FSA (https://github.com/k2-fsa/k2), which seems to be aiming to similar goals or features.



**Limitations:**


The authors have analysed the limitations of their proposal in terms of training time and cost which is one of the most important questions a model such as the presented one raises.

**Strengths And Weaknesses:**

 The paper seems relevant but mainly  for the ASR community, which could be an strength or weakness depending on the readers background.

The proposed GNAT or general model seems as an application of classical ASR framework to the deep learning (DL) models that are currently the state of the art. The classical formalisms, well known from HMM-based AM models,  are introduced to generalize many DL models into the wFSA/T formalism similarly as done in ref [16].

The authors relate their proposed model with standard DL models. They provide examples of their proposed merged framework as well as mapping  well-known standard architectures within it.
They also  argue that the globally normalized models improve streaming models by fixing the label bias problem providing some empirical evidence.

However, there are many questions and experiments missing for a full understanding of the main contribution of the paper, which is the globally normalized RNN-T like models for ASR. While from the engineering perspective it is clear the paper was a huge internal effort, the main contribution of an efficient way to compute a globally normalized model in a wFSA framework could have been analyzed further.
It is not clear to which extend the problems on label bias are major for the streaming ASR models or in which circumstances. For instance, there are encoders with small look-ahead in the literature or dual [1] streaming/non-streaming encoders that could also solve the label bias problem, etc -- see below--. It is not clear how this 2 potential solutions will interact for instance.


[1] Dual-mode ASR: Unify and Improve Streaming ASR with Full-context Modeling ( https://openreview.net/forum?id=Pz_dcqfcKW8 )

---

> ### Author Response · Authors · 2022-08-02
> **Response to Reviewer wc8G (Part I)**
>
> We once again thank the reviewer for their time & valuable feedback. Here we address their questions & concerns.
>
> > The paper seems relevant but mainly for the ASR community, which could be an strength or weakness depending on the readers background.
>
> Please consider that sequence prediction has seen wide applications not only in ASR but also TTS,NLP, image/video generation, etc. The discussion of the relationship between globally & locally normalized models is thus applicable to a much wider audience. The modular design of GNAT means it can be applied to many problems outside ASR, especially those that also need to meet stringent latency requirements and thus require streaming. We are thus hopeful that the diverse NeurIPS community would see value in our work beyond the improvement in ASR modeling.
>
> > However, there are many questions and experiments missing for a full understanding of the main contribution of the paper, which is the globally normalized RNN-T like models for ASR.
>
> We would like to bring the reviewer’s attention to the highlights of our contribution at the end of introduction. We believe that our contribution is not only the “globally normalized RNN-T like models for ASR”, but a general framework which,
> - Encompasses ALL the common models in ASR (CE, CTC, LAS, RNNT, HAT), and
> - Furthermore extends to their global normalization counterparts which shows significant benefit for streaming settings.
>
> All the experiments are designed to establish these points and we think they are supporting this claim. Having said that, there are many directions that we would like to explore in the future by expanding the current work.
>
> > While from the engineering perspective it is clear the paper was a huge internal effort, the main contribution of an efficient way to compute a globally normalized model in a wFSA framework could have been analyzed further. It is not clear to which extend the problems on label bias are major for the streaming ASR models or in which circumstances. For instance, there are encoders with small look-ahead in the literature or dual [1] streaming/non-streaming encoders that could also solve the label bias problem, etc -- see below--. It is not clear how this 2 potential solutions will interact for instance.
>
> > [1] Dual-mode ASR: Unify and Improve Streaming ASR with Full-context Modeling ( https://openreview.net/forum?id=Pz_dcqfcKW8 )
> We devoted whole section 2 to discuss the streaming problem our paper is addressing and why locally normalized models are insufficient to address the modeling problem. Please see our comments regarding e.g. lookahead below.
>
> > experiments on how the performance changes with the encoder look-ahead
> We omitted experiments with lookahead mainly for two reasons:
>
> This paper focuses on improving the streaming model without introducing lookahead delays in order to meet stringent latency requirements. Towards this goal, we aimed to show in our experiments  that our best globally normalized streaming model fills nearly 50% of the performance gap between previously best locally normalized streaming and non-streaming models.
> We believe that adding lookahead (delay, or right context) to improve streaming models is already explored in literature (for example references a and b below). We therefore chose to focus on zero lookahead models due to the space & time limitations. We note that the performance of non-streaming models, which are instances of full lookahead, are still better than streaming models, for both local and global models. Therefore, we believe adding lookahead to globally normalized streaming models can still give us gains, a direction that we are very interested in exploring in the future.
>
> [a] Albert Zeyer, Ralf Schlüter, and Hermann Ney, “Towards online-recognition with deep bidirectional lstm acoustic models.,” in Interspeech, 2016, pp. 3424–3428.
>
> [b] Niko Moritz, Takaaki Hori, and Jonathan Le Roux, “Unidirectional neural network architectures for end-to-end automatic speech recognition.,” in INTERSPEECH, 2019, pp. 76–80.
>
> (to be continued)

---

> > ### Author Response · Authors · 2022-08-02
> > **Response to Reviewer wc8G (Part II)**
> >
> > > analysis of different approaches for approximating the denominator sum, e.g. n-best, lattices ,,,
> >
> > The majority of globally normalized models in literature use approximation methods like running the decoder (to obtain n-best or lattice) and using that for computing the denominator. One major contribution of our paper is proposing models where we can compute the exact value of the denominator. Since we can already carry out the exact computation, we did not explore methods of approximation as of yet.
> >
> > > analysis/experiments on the conditions for which the label bias is a major problem by for instance what would happen on different datasets or what would happen if we have noisy training data , ... etc,
> >
> > The label bias problem is not a problem specific to a particular dataset, rather it is a well studied problem, caused by local normalization, that has been seen to apply very widely. We refer the reviewer to Section 3 of [2] (Andor et al, 2016) for a concise yet comprehensive discussion on this topic. Also as mentioned in our experiment section, SpecAugment is used for training (line 262). So indeed the current set of results are all from models trained using noisy training data.
> >
> > > the paper uses a very small vocabulary of 30 characters, should the same conclusions arise with word-level models composed of thousands of words/word-pieces ?
> >
> > On theoretical grounds, we certainly believe that the main result on the benefit of global normalization over local normalization for streaming models will translate to different types of units. We are actively working on scaling up GNAT to use larger context size and vocabulary (please see our response to Reviewer j8Qd on scalability for details).
> >
> > > Finally, despite the novelty of the globally normalized model, the paper seems related to K2 FSA (https://github.com/k2-fsa/k2), which seems to be aiming to similar goals or features.
> >
> > We would like to note that k2 is not a specific model like GNAT, but a general library for implementing ASR models. That being said, although our model can presumably be implemented with k2, we implemented our algorithms from scratch to run on TPUs, which is why k2 was not used. We have revised the first paragraph of Section 3 to clarify on this topic.

---

### Official Review · Reviewer_j8Qd · 2022-07-11

**Rating:** 6
**Confidence:** 4
**Soundness:** 3 good
**Presentation:** 2 fair
**Contribution:** 3 good

**Summary:**

This paper introduces the Globally Normalized Autoregressive Transducer (GNAT) for addressing the label bias problem in streaming speech recognition.
An exact computation of the denominator for the sequence-level normalization is realized in the WFSA framework. The WFSA based modular framework potentially can encompass some common neural speech recognition models.
Experiments are conducted on Librispeech and show that GNAT can close 50% of the WER gap between streaming and non-streaming ASR.

**Questions:**

see above

**Limitations:**

see above

**Strengths And Weaknesses:**

-Strengths

The paper is well organized and written. The content is intensive.

Based on the solid theory of WFSA, the GNAT model is new and successfully developed.

It is well known that the label bias problem suffered by locally normalized models can be naturally overcome by globally normalized models. It is nice to see that this paper provides a new insight that such advantage of globally normalized models in overcoming label bias can bring evident performance improvement in streaming speech recognition, with thorough empirical validations.

-Weaknesses

However, I have the following concerns.

I) GNAT seems to be only feasible for small number of units.

The difficulty in training globally normalized models is the computation of the denominator (aka partition function in random field terminology). The core contribution of this paper is that based on theory of WFSA, it realizes an efficient algorithm for its exact computation, but still only for small number of units (28 graphemes, |\Sigma|=32, in this paper) and limited context dependency.

Current state-of-the-art ASR systems often use word-pieces, which usually contain thousands of units. It is not clear whether the proposed algorithm can be applied to such setting. (Line 247-258 only discusses the complexity w.r.t. n (the order of context dependency))

Line 76-77: "However with our proposed modular framework, globally normalized model training can be made practical with careful modelling choices on modern hardware."

This claim only works for small number of units.

II) Imprecise comments about locally and globally normalized models.

Line 78-79: For non-streaming settings, [33] shows that locally and globally normalized models express the same class of conditional distributions P(z|x).

Line 319-320: the equal expressiveness of globally and locally normalized models under non-streaming setting [33].

Line 245-246: all these models are non-streaming, where as explained multiple times in our paper, global normalization and local normalization are equally expressive.

In fact, the paper of [33] does not draw such a strong conclusion "the equal expressiveness of globally and locally normalized models"; it only compares weighted context-free grammars (WCFGs) and probabilistic context-free grammar (PCFG) - the two very particular models, and in the non-parametric setting.

Basically, globally normalized models are also well known as random fields or undirected graphical models or Markov networks; locally normalized models as directed graphical models or Bayesian networks. It is well documented in Section 4.5 from [a] about the relationship between the two classes of models.

Also for your information, it is said in [b] that:
"...the label bias problem implies that globally normalized models can be strictly more expressive than locally normalized models."

III) The introduction to prior work on Globally Normalized Models should be improved. A number of statements are imprecise.

Line 234-241: the paper claims 4 differences between GNAT and MMI.
The third difference is imprecise: the exact computation of the denominator is also taken in the lattice-free MMI [31].

Line 242-246: These models can be seen as special cases of MMI thus all the differences between MMI and GNAT model applies here as well. ...

The CTC-CRF model [37,39] should not be classified as special cases of MMI. In fact, this work is closely related to the CTC-CRF model [37,39]. Similar to CTC-CRF, GNAT is a globally normalized model and is directly trained via conditional maximum likelihood, without any need for initialization or special regularization techniques. A main novelty in GNAT is that it uses WFSA based parameterization, which is more flexible.

It is important to establish clearer concepts (e.g. differing model definition and model optimization) and give the readers more precise picture.

IV) Imprecise statements about the label bias problem in streaming and non-streaming settings (Section 2.1 and 2.2).

It is said in this paper: In non-streaming settings, score w(z_i | z_{<i}, x) can be directly expressed as probability p(z_i | z_{<i}, x). In streaming settings, \prod w(z_i | z_{< i}, x_{<t(i)) is in general not equal to \prod p(z_i | z_{<i}, x), so there is label Bias problem in streaming settings.

However, in non-streaming settings, modeling with w(z_i | z_{<i}, x) (or say, via directed sequential models in general) inherently has the problem of label bias [a,b,c]. To be precise, the label bias problem is more severe in streaming settings, and less severe in non-streaming settings.

V) About M-gram context dependency and the RNN-T baseline.
This part of content is hard to follow.

"M is equal to the length of the longest feature sequence in the training data (1961) and for label frame dependent M is equal to 1961 + 384."
No complexity analysis for computing the denominator in this case.

More introduction to the RNN-T baseline is needed. Is it implemented as a GNAT?

Minors:
Line 594: Sigma -> \Sigma

[a] D. Koller and N. Friedman. Probabilistic graphical models: principles and techniques. MIT press, 2009.

[b] Andor, et al., "Globally Normalized Transition-Based Neural Networks", ACL, 2016.

[c] Wiseman, et al., "Sequence-to-sequence Learning as Beam-Search Optimization", EMNLP, 2016.

---

> ### Author Response · Authors · 2022-08-02
> **Response to Reviewer j8Qd (Part I)**
>
> We once again thank the reviewer for their time & valuable feedback. Here we address their questions & concerns.
>
> > I) GNAT seems to be only feasible for small number of units.
>
> > The difficulty in training globally normalized models is the computation of the denominator (aka partition function in random field terminology). The core contribution of this paper is that based on theory of WFSA, it realizes an efficient algorithm for its exact computation, but still only for small number of units (28 graphemes, |\Sigma|=32, in this paper) and limited context dependency.
> Current state-of-the-art ASR systems often use word-pieces, which usually contain thousands of units. It is not clear whether the proposed algorithm can be applied to such setting. (Line 247-258 only discusses the complexity w.r.t. n (the order of context dependency))
>
> > Line 76-77: "However with our proposed modular framework, globally normalized model training can be made practical with careful modelling choices on modern hardware."
>
> > This claim only works for small number of units.
>
> We are actively working on enabling GNAT to use larger context sizes and larger vocabulary sizes right now, and we are confident that scaling up is highly feasible. We thank the reviewer for their interest in this!
>
> Without getting into too much detail, let’s analyze the time & space complexity of training & inference in GNAT, using the frame dependent alignment lattice as an example. Let T be the number of input frames, Q be the number of states in the context dependency, and V be the vocabulary size. It turns out that during both training & inference, our model uses O(T Q V) time (or more precisely “work” in a parallel computing setting), and O((T+V) Q) space. With the complexity in mind, we would note the following:
>
> - We have plenty of memory headroom: Q is currently just above 1000, and thus for T=1024 and V=32, our algorithm is just using a few megabytes of memory not counting the memory used for storing parameters.
> - Scaling up by Q & V can be easily parallelized: the vast majority of computation affected by Q & V is in computing the arc weights, which is an easily parallelizable task. For example, if we can hold Q constant, increasing V 32-fold to 1024 is actually very practical (using V=32 is actually underutilizing the computing power of modern accelerators).
>
> The key insight is that we need a way to control Q independent of V: currently the n-gram context dependency doesn’t allow us to do that. However, there are many ways to achieve this while keeping the context dependency general enough to still accept $\Sigma^*$: for example, we can use a n-gram topology with backoff, or merge states by other criteria such as context state distance according to an RNN language model.
>
> (to be continued)

---

> > ### Author Response · Authors · 2022-08-02
> > **Response to Reviewer j8Qd (Part II)**
> >
> >
> > > II) Imprecise comments about locally and globally normalized models.
> >
> > > Line 78-79: For non-streaming settings, [33] shows that locally and globally normalized models express the same class of conditional distributions P(z|x).
> >
> > > Line 319-320: the equal expressiveness of globally and locally normalized models under non-streaming setting [33].
> >
> > > Line 245-246: all these models are non-streaming, where as explained multiple times in our paper, global normalization and local normalization are equally expressive.
> >
> > > In fact, the paper of [33] does not draw such a strong conclusion "the equal expressiveness of globally and locally normalized models"; it only compares weighted context-free grammars (WCFGs) and probabilistic context-free grammar (PCFG) - the two very particular models, and in the non-parametric setting.
> > Basically, globally normalized models are also well known as random fields or undirected graphical models or Markov networks; locally normalized models as directed graphical models or Bayesian networks. It is well documented in Section 4.5 from [a] about the relationship between the two classes of models.
> >
> > > Also for your information, it is said in [b] that: "...the label bias problem implies that globally normalized models can be strictly more expressive than locally normalized models."
> >
> > We thank the reviewer for paying close attention to our analysis---it is of enormous help to improve the quality of our paper! Indeed the relationship between locally & globally normalized models is quite nuanced, in both directions. The discussions on this topic in the literature largely focused on when a locally normalized model is strictly less expressive (just like your quote from Andor et al, 2016 or the PGM book). Our goal in the second half of Section 2.2 was to discuss the reverse direction: when a globally normalized model is not strictly more expressive than a locally normalized counterpart. This topic in our opinion has not received enough attention: under a mild condition that holds for all the neural ASR models that we know of, non-streaming globally normalized models and their locally normalized counterparts are actually equally expressive, which is why past work on globally normalized neural ASR did not achieve an improvement in word error rate when compared against a locally normalized counterpart.
> >
> > Having explained our motivation, we do agree that we could have done a better job explaining this in the paper. [33] (Smith & Johnson, 2007) is arguably not the best source for supporting our claim, even though the technique needed for our proof was there. We have revised the parts of the paper pointed out by the reviewer, and included a proof for our claim as a new appendix.
> >
> > From our experiences working with many colleagues, explaining the nuances of sequence modelling using graphical models is often not effective. Despite graphical models being indeed a powerful tool, sequence modelling is a very concrete special case where many aspects of the theory can be explained explicitly and clearly without background knowledge on graphical models. In order to reach a larger audience, we strive to be self-contained in this aspect.
> >
> > > Line 234-241: the paper claims 4 differences between GNAT and MMI. The third difference is imprecise: the exact computation of the denominator is also taken in the lattice-free MMI [31].
> >
> > We thank the reviewer for pointing out a possible source of confusion, and we have revised our discussion in this part.
> >
> > Please note that the set of paths included in GNAT’s exact denominator computation at training time is identical to the set of possible paths for the given input utterance at inference time, weighted the same way. Although the scope of summing for the denominator is nominally the same for MMI models, all past MMI models have employed various types of approximation to compute the denominator, e.g. summing only a subset of paths from a pruned lattice, or using a smaller language model at training time and switching to a larger one at inference time.
> >
> > While lattice-free MMI does not prune the lattice, it
> > - Uses different LMs between training time (a 4-gram phone level LM, Section 2.3 of [31] (Povey et al, 2016)) and inference time (a trigram LM, presumably word-based, second paragraph of Section 3 of [31] (Povey et al, 2016));
> > - Splits the utterance into small chunks (typically 1.5 seconds long, see Section 2.5 of [31] (Povey et al, 2016)). Therefore the utterance level denominator is still approximate in lattice-free MMI.
> >
> > (to be continued)

---

> > > ### Author Response · Authors · 2022-08-02
> > > **Response to Reviewer j8Qd (Part III)**
> > >
> > >
> > > > Line 242-246: These models can be seen as special cases of MMI thus all the differences between MMI and GNAT model applies here as well. ...
> > >
> > > > The CTC-CRF model [37,39] should not be classified as special cases of MMI. In fact, this work is closely related to the CTC-CRF model [37,39]. Similar to CTC-CRF, GNAT is a globally normalized model and is directly trained via conditional maximum likelihood, without any need for initialization or special regularization techniques. A main novelty in GNAT is that it uses WFSA based parameterization, which is more flexible.
> > >
> > > Please see below a list of characteristics of CTC-CRF that makes it much closer to MMI than GNAT:
> > >
> > > - Score formulation (Section 3.3 of [37] (Xiang and Ou, 2019)): CTC-CRF’s potential function has two components, the sum of the “acoustic model” log-probabilities $\sum_t \log p(\pi_t | \mathbf{x})$, and the log-probability from a language model $\log p(\mathbf{l})$. This is not really that different from the classic MMI score formulation. In contrast, the weight functions we use in GNAT do not impose such a factorization, and are thus much more powerful.
> > > - Fixed LM at training time (Section 3.3 of [37] (Xiang and Ou, 2019)): $\log p(\mathbf{l})$ in CTC-CRF comes from an external “n-gram denominator LM of labels” (i.e. characters or phones), fixed during training, like all previous MMI models. In contrast, all the model parameters are trained together in GNAT.
> > > - Different LM at inference time (Section 3.4 of [37] (Xiang and Ou, 2019)): At inference time, a word level LM is used instead, replacing the training time LM, like many previous MMI models. As we have discussed in our previous response regarding the exact computation of the denominator, this mismatch means the denominator computed at training time by CTC-CRF is based on an LM different from the one used at inference time, and thus not exact. In contrast, because GNAT doesn’t use an external LM, the computation of the denominator sums over the same weighted paths the model will see at inference time.
> > >
> > > We omitted a detailed comparison with CTC-CRF due to space limitations, because the comparisons above draw direct parallels to the comparisons between MMI and GNAT we have already made in the previous paragraph in our paper. We thank the reviewer for bringing this up and we have added some revisions.
> > >
> > > > IV) Imprecise statements about the label bias problem in streaming and non-streaming settings (Section 2.1 and 2.2).
> > > It is said in this paper: In non-streaming settings, score w(z_i | z_{<i}, x) can be directly expressed as probability p(z_i | z_{<i}, x). In streaming settings, \prod w(z_i | z_{< i}, x_{<t(i)) is in general not equal to \prod p(z_i | z_{<i}, x), so there is label Bias problem in streaming settings.
> > >
> > > > However, in non-streaming settings, modeling with w(z_i | z_{<i}, x) (or say, via directed sequential models in general) inherently has the problem of label bias [a,b,c]. To be precise, the label bias problem is more severe in streaming settings, and less severe in non-streaming settings.
> > >
> > > Since the reviewer is discussing locally normalized models, we believe this comment relates to Section 2.1 alone, and more particularly, the first paragraph. Section 2.1 is comparing the capability of locally normalized models between non-streaming and streaming settings in the limit (i.e. with enough model capacity). We have revised our text to make this clearer.
> > >
> > > More specifically, in the first paragraph of Section 2.1, we are merely saying that with enough model capacity, a non-streaming locally normalized model can express the true posterior distribution $P(\mathbf{z} | \mathbf{x})$. Using graphical model terms, we are saying that the following Bayesian network can represent any conditional distribution $P(\mathbf{z} | \mathbf{x})$:
> > > - There is a single node x representing the input, and nodes {z_1, z_2, ...} representing the output;
> > > - There is a directed edge from x to all z_i, and a directed edge from z_i to z_j for any i < j.
> > >
> > > This is however very different from saying that there is no label bias in any non-streaming locally normalized models, which might be the reviewer’s interpretation of this paragraph. Whether a model exhibits the symptom of label bias depends on many other factors, such as the true data distribution, the actual model capacity, and independence assumptions (if any) in the model. This is also why we discuss the situation in the limit instead of under a particular model parameterization.
> > >
> > > (to be continued)

---

> > > > ### Author Response · Authors · 2022-08-02
> > > > **Response to Reviewer j8Qd (Part IV)**
> > > >
> > > > > V) About M-gram context dependency and the RNN-T baseline. This part of content is hard to follow.
> > > >
> > > > > "M is equal to the length of the longest feature sequence in the training data (1961) and for label frame dependent M is equal to 1961 + 384." No complexity analysis for computing the denominator in this case.
> > > > More introduction to the RNN-T baseline is needed. Is it implemented as a GNAT?
> > > >
> > > > Sorry about the confusion caused here. We meant to use “M-gram” to denote the RNN-T model trained with an unlimited history. We realize that this might be confusing and have revised our description. To answer the reviewer’s final question: We used an existing implementation of RNN-T for the baseline experiments.

---

### Official Review · Reviewer_KkKb · 2022-07-12

**Rating:** 7
**Confidence:** 5
**Soundness:** 4 excellent
**Presentation:** 3 good
**Contribution:** 3 good

**Summary:**

This paper proposes a globally normalized speech recognition model. The model consists of two finite-state acceptors, one that provides context (reminiscent to a language model), and the other that provides the alignment from frames to labels. The paper argues that the performance of streaming is worse because of the label bias problem, and that training models with global normalization alleviates the problem. Experiments show that, in the streaming setting, global normalization generally performs better than local normalization.

**Questions:**

The FSA looks deceptively small, but the FSA actually describes paths of all possible lengths. It is probably worth noting this in the paper that once these FSAs are composed, the final FSA could have exponentially many edges.

The label bias seems to be a mismatch between training and test conditions, not so much about global versus local normalization. I do agree that globally normalized models would consider those partial paths during training, but wouldn't a locally normalized trained in streaming mode alleviates the problem too?

> ... by minimizing the negative log-conditional-likelihood loss E_{P(x, y)}[...] = ...

P(x, y) is not defined.

> It is worth noting that any locally normalized model is trivially a globally normalized because Z(x) = 1 in this case.

It is not as trivial as it seems, and boils down to the distinction between Bayesian networks and Markov random fields. The community has been aware of this ever since the existence of discriminative training. I believe the first to discuss this in the end-to-end setting is Tang et al., 2017.

Tang et al., End-to-end neural segmental models for speech recognition, IEEE Journal of Selected Topics in Signal Processing, 2017

> ... even though our algorithms cannot be directly implemented using existing toolkits such as OpenFst [1] or Kaldi [30].

In principle, we could. I'm not entirely sure what's the intention here.

> ... under the log semiring can be viewed as the unnormalized negative log conditional probability P_\theta(y | x) = \frac{\exp(-W(A_{\theta, x}(y)))}{...}

Shouldn't W(A_{\theta, x}(y)) be just A_{\theta, x}(y)?

> Our separation of the weight function from the automaton topology allows an arbitrarily complex, non-linear weight function to model the dependency among alignment states, context states, and output label, which is impossible with composition cascades.

I believe this was proposed in Tang et al., 2015, called structured composition. There could be earlier papers.

Tang et al., Discriminative segmental cascades for feature-rich phone recognition, ASRU, 2015

> The transcript truth is used without any processing and tokenized ...

Ground truth? Or just transcript?

**Limitations:**

The societal impact is the same as any other ASR work.

**Strengths And Weaknesses:**

The major strength is the formalism to put several ASR models under the same framework. The examples in Figure 1 are also very helpful for understanding the framework.

Overall, this is a solid paper. There are several sections that are a bit notation heavy, but that's the nature of describing FSAs.

---

> ### Author Response · Authors · 2022-08-02
> **Response to Reviewer KkKb (Part I)**
>
> We once again thank the reviewer for their time & valuable feedback. Here we address their questions & concerns.
>
> > The FSA looks deceptively small, but the FSA actually describes paths of all possible lengths. It is probably worth noting this in the paper that once these FSAs are composed, the final FSA could have exponentially many edges.
>
> We are not entirely sure which FSA the reviewer is referring to, so we would instead examine the asymptotic sizes of all FSAs in Figure 1, and the size of the induced recognition lattice:
>
> - The n-gram context dependency has $O(|\Sigma|^n)$ states, and $O(|\Sigma|^{n+1})$ arcs (see “Challenges” in Section 5). This is thus exponential in the context size $n$, but note $\Sigma$ and $n$ are both fixed during a single experiment.
> - Let $T$ be the number of input frames,
>     - The frame dependent alignment lattice has $O(T)$ states and $O(T |\Sigma|)$ arcs.
>     - The $k$-constrained label and frame dependent alignment lattice has $O(k T)$ states and $O(k T |\Sigma|)$ arcs.
>     - The label dependent alignment lattice has $O(l(T))$ states and $O(l(T) |\Sigma|)$ arcs.
> - The induced recognition lattice’s topology, like any other FSA intersection, has a number of states & arcs polynomial to its input sizes. For example, when using the frame dependent alignment lattice, the recognition lattice will have $O(T |\Sigma|^n)$ states, and $O(T |\Sigma|^{n+1})$ arcs.
> - It is also worth noting that the alignment lattice is acyclic and dependent on the input length (see Section 3.3). Therefore the recognition lattice conditioned on the input describes not “paths of all possible lengths”, just the ones allowed by the alignment lattice, which is bounded due to the alignment lattice being acyclic.
>
> In summary, the asymptotic sizes of our FSAs are comparable to existing approaches. We omitted the discussions above in the paper due to space limitations.
>
> > The label bias seems to be a mismatch between training and test conditions, not so much about global versus local normalization. I do agree that globally normalized models would consider those partial paths during training, but wouldn't a locally normalized trained in streaming mode alleviates the problem too?
>
> For all our experiments, the training & testing conditions match with respect to streaming: if the encoder is trained in streaming mode, we only test the model in streaming mode; if the encoder is trained in non-streaming mode, we only test the model in non-streaming mode. As we can see from Table 1a, there is quite a significant accuracy gap between streaming and non-streaming baselines despite matching training & testing conditions.
>
> Label bias in the context of streaming sequence prediction using locally normalized models is a symptom of such locally normalized models’ inability to express some conditional distributions that a globally normalized model parameterized in the same way can. We refer the reviewer to Section 3 of [2] (Andor et al, 2016) for a concise yet comprehensive discussion on this topic.
>
> >> ... by minimizing the negative log-conditional-likelihood loss E_{P(x, y)}[...] = ...
>
> > P(x, y) is not defined.
>
> Thank you for your feedback! We meant the true data distribution, and we have clarified this in our revised submission.
>
> (to be continued)

---

> > ### Author Response · Authors · 2022-08-02
> > **Response to Reviewer KkKb (Part II)**
> >
> > >>... even though our algorithms cannot be directly implemented using existing toolkits such as OpenFst [1] or Kaldi [30].
> >
> > > In principle, we could. I'm not entirely sure what's the intention here.
> >
> > We mainly intended to draw a distinction between using WFSA as a formal framework and using existing toolkits. This has been a major confusion for some readers in the past. We have revised the sentence to “our algorithms are implemented from scratch to run on TPUs, without using existing toolkits”.
> >
> > >> ... under the log semiring can be viewed as the unnormalized negative log conditional probability P_\theta(y | x) = \frac{\exp(-W(A_{\theta, x}(y)))}{...}
> >
> > > Shouldn't W(A_{\theta, x}(y)) be just A_{\theta, x}(y)?
> >
> > Thank you for catching this! We have revised the relevant part.
> >
> > >> Our separation of the weight function from the automaton topology allows an arbitrarily complex, non-linear weight function to model the dependency among alignment states, context states, and output label, which is impossible with composition cascades.
> >
> > > I believe this was proposed in Tang et al., 2015, called structured composition. There could be earlier papers.
> > Tang et al., Discriminative segmental cascades for feature-rich phone recognition, ASRU, 2015
> >
> > Thanks for bringing this to our knowledge! We have added a citation. It is not our intention to claim originality for this idea, thus we have revised “our separation” to “the separation”. At the same time, we would note that in addition to separating the weight function from the topology, how exactly the weight function is designed also plays a crucial role in model accuracy as we can see from Table 1c.
> >
> > >> The transcript truth is used without any processing and tokenized ...
> >
> > > Ground truth? Or just transcript?
> >
> > Thank you for your feedback! We have revised this to “ground truth transcription”. Technically speaking, “transcript” can be either from a human or a machine, which is why we made the distinction.

---

### Author Response · Authors · 2022-08-02
**Summary**

We thank all the reviewers for their time and valuable feedback. We are happy to see that our main contributions are recognized by all the reviewers, which we shall highlight below:

- Addressing the label bias problem in streaming ASR is one of the major contributions of our paper and it was encouraging to see that reviewers realized it: “It is nice to see that this paper provides a new insight that such advantage of globally normalized models in overcoming label bias can bring evident performance improvement in streaming speech recognition, with thorough empirical validations.” (reviewer j8Qd), “This work mitigates the performance gap between the streaming and non-streaming systems by solving the label bias problem.” (reviewer tQ4B).

- Reviewers also recognized the novelty and theoretical significance of our paper: “Overall, this is a solid paper. ” (reviewer KkKb), “Based on the solid theory of WFSA, the GNAT model is new and successfully developed.” and “this paper provides a new insight” (reviewer j8Qd), “a unified wFST theoretical framework to unify many of the common ASR architectures” (reviewer wc8G), "Theoretical soundness under WFSA framework” and “Moreover, this work is unlike other works, which simply tune the structure of networks. It is concrete in both theory and implementation.” and “It is a novel idea.” (reviewer tQ4B).

- We tried to carefully design through experiments comparing different components of the GNAT model with the focus on the importance of the global normalization for improving streaming models which was acknowledged by the reviewers: “with thorough empirical validations.” (reviewer j8Qd), “The results are interesting” (reviewer wc8G), “Moreover, this work is unlike other works, which simply tune the structure of networks. It is concrete in both theory and implementation. The experimental results in streaming ASR through the proposed global normalization significantly close more than 50% of the WER to a non-streaming system.” (reviewer tQ4B).

- Our proposed modular framework allows interpretation of many common neural ASR models and allows extending them to new ones which is also observed by several reviewers: “The major strength is the formalism to put several ASR models under the same framework.” (reviewer KkKb), “they propose a unified wFST theoretical framework to unify many of the common ASR architectures” (reviewer wc8G).

- While we are still actively improving the text, we appreciate the reviewers' overall positive evaluation and remarks about the current state of our paper’s presentation: “The paper is well organized and written. The content is intensive.” by reviewer j8Qd.

We once again thank the reviewers, and we shall respond to each reviewer’s questions & concerns individually below.

---

### Author Response · Authors · 2022-08-10
**Follow up**

Dear reviewers,
We are happy to respond to any additional questions & comments. We hope our continued effort to improve the submission  strengthens the generally favorable view we believe we have heard from our reviewers, addresses many of the concerns they have pointed out and leads to its full acceptance.

---

### Meta-Review · Area_Chair_cEKH · 2022-08-26

**Recommendation:** Accept
**Confidence:** Less certain

**Metareview:**

Reviewers acknowledge that this paper has good contributions, including solving label bias problem, mitigating the performance gap between the streaming and non-streaming systems, and proposing a unified WFST theoretical framework, which should have value to the ASR community. However, reviewer j8Qd reported that he/she did not receive the notifications of the author response. He/she then posted new problems on the revised paper on Aug. 26, while the authors cannot respond. The problems are mainly on the comparison with prior works on the mismatching training and inference. The reviewer argued that there are on imprecise comments/claims. Given this situation, I give a recommendation of acceptance but strongly ask the authors to carefully look into these problems and address them with full effort.

**Award:**

No

---

### Decision · Program_Chairs · 2022-09-14

Accept